# GeoDetect: Geometric Adversarial Detection for VLPs

## Abstract

Vision-language pre-trained models (VLPs) are widely used in real-world applications. However, they remain vulnerable to adversarial attacks. Although adversarial detection methods have demonstrated success in single-modality settings (either vision or language), their effectiveness and reliability in multimodal models such as VLPs remain largely unexplored. In this work, we investigate the embedding spaces of VLPs and find that the image embedding space exhibits anisotropy. Our theoretical analysis shows that this *anisotropic structure increases the separation between clean and adversarial examples (AEs) in the embedding space.* Specifically, we demonstrate that AEs consistently exhibit greater expected distances to randomly sampled points than their clean counterparts, indicating that adversarial perturbations tend to push inputs out of manifold regions. Building on these insights, we propose GeoDetect, which leverages these off-manifold deviations to identify AEs. Through comprehensive evaluations, we show that our approach reliably detects adversarial attacks across various VLP architectures, including but not limited to CLIP, providing a robust and practical approach to improving the safety and reliability of these models.

## 1 Introduction

Vision-language pre-trained models (VLPs) enable the understanding of both visual and textual data by learning joint representations of multimodal inputs. This capability makes them highly effective for tasks requiring a deep understanding of images and text. VLPs have achieved state-of-the-art results across various multimodal tasks (Yin et al., 2023a; Xu et al., 2023; Gandhi et al., 2023), including image-text retrieval (Chen et al., 2020a), visual question answering (Lu et al., 2019), and zero-shot classification (Radford et al., 2021). Despite their remarkable success, VLPs remain vulnerable to adversarial examples (AEs) (Zhang et al., 2022a; Schlarmann & Hein, 2023), raising concerns about their robustness in real-world, safety-critical applications.

Recent research has explored adversarial training as a strategy to enhance the zero-shot robustness of VLPs (Mao et al., 2022; Wang et al., 2024a; Schlarmann et al., 2024). However, adversarial training is computationally expensive (Madry et al., 2017; Wang et al., 2020) and often involves a trade-off between model performance and robustness (Zhang et al., 2019; Tsipras et al., 2019). Detecting AEs presents a more flexible alternative by allowing the model to identify and reject potentially harmful queries, rather than attempting to provide reliable outputs for all inputs.

While existing work has been proposed for detecting AEs in unimodal models (Feinman et al., 2017; Lee et al., 2018; Ma et al., 2018; Sotgiu et al., 2020; Kherchouche et al., 2020; Aldahdooh et al., 2023), it remains uncertain whether these approaches can effectively generalize to VLPs, which integrate two interacting modalities. Most existing research has focused on the detection of adversarial images in classification models. In contrast to standard classifiers trained with cross-entropy loss to predict discrete labels, VLPs are optimized to align image and text representations within a shared embedding space using contrastive learning objectives (Radford et al., 2021). Recent findings by Schlarmann et al. (2024) demonstrate that CLIP embeddings experience significant distortion under adversarial attack, as evidenced by substantial shifts in the embedding space. A recent study by Zhang et al. (2024c) proposed Prompt-based Irrelevant Probing (PIP), a task-specific detection method for visual question answering (VQA) that analyzes attention responses to irrelevant probe questions. However, PIP's applicability is limited to architectures employing explicit cross-attention mechanisms, and its

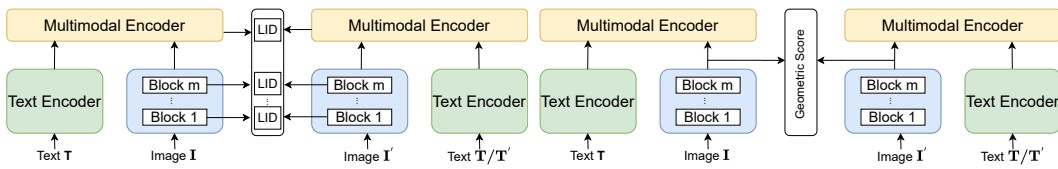

(a) Extraction of LID scores.      (b) Extraction of $k$-NN, Mahalanobis, and KDE.

Figure 1: Pipeline of geometric score extraction for GeoDetect.

reliance on question-conditioned attention constrains it exclusively to VQA tasks. Thus, the absence of a comprehensive and theoretically grounded investigation into adversarial detection for VLPs leaves a critical gap in understanding their vulnerabilities and robustness. More importantly, no theoretical understanding currently explains the nature of AEs in VLPs.

In this work, we investigate the intrinsic properties of VLPs by revisiting the anisotropic nature of CLIP's embedding space, as observed in prior work (Liang et al., 2022; Levi & Gilboa, 2024), and extending this analysis to other VLPs. While existing studies primarily focus on CLIP, we show that this property extends to a broader type of VLPs, forming the foundation for our theoretical assumptions. Anisotropy indicates that representations are unevenly dispersed across embedding dimensions, creating dense and sparse directions. We leverage this property to formulate our central theoretical contribution: *AEs explore off-manifold regions of the embedding space, resulting in a greater expected distance between an AE and a random clean example, compared to the distance between the unperturbed version to the same random clean example.* This motivates us to investigate the geometric properties surrounding data representations, through which we uncover fundamental differences between the regions occupied by adversarial and clean examples.

Building on our theoretical insights, we propose GeoDetect, an effective method for detecting AEs in VLPs. GeoDetect extracts deep representations from VLP encoders and applies classical geometric metrics to compute detection scores, including Local Intrinsic Dimensionality (LID) (Houle, 2013), k-Nearest Neighbours distance (k-NN), Mahalanobis distance (McLachlan, 1999), and Kernel Density Estimation (KDE). These scores are then used for logistic regression or threshold-based detection of AEs. An overview of GeoDetect is illustrated in Figure 1. Figure 1a presents adversarial image detection using LID, while Figure 1b illustrates detection using the other three studied methods, $k$-NN, Mahalanobis distance, and KDE. A difference is that LID operates layer-wise, evaluating the outputs of both multimodal layers and other intermediate layers, while the other three methods operate on the output of the image encoder, making LID more sensitive to perturbations across the multimodal encoder. Built upon solid theoretical foundations, GeoDetect offers a cost-effective and robust method for ensuring the safety of VLPs.

The main contributions can be summarized as follows:

- We analyze the anisotropic structure of VLP embedding spaces and, building on this, theoretically demonstrate that AEs tend to lie in off-manifold regions, resulting in larger geometric deviations from clean reference points.

- We introduce GeoDetect, a novel model-agnostic detection method that employs geometric discrepancies through a family of metrics to identify AEs in VLPs across different downstream tasks (e.g., zero-shot classification and retrieval).

- We comprehensively validate the effectiveness of GeoDetect across various VLP architectures and attacks, achieving consistently high AUC scores, all without requiring fine-tuning. This makes GeoDetect both robust and lightweight compared to existing defense methods.

## 2   RELATED WORK

**Vision-Language Pre-Trained Models.** Vision-language representation learning outperforms visual representation learning across a wide range of tasks. For instance, CLIP uses a contrastive objective (i.e., InfoNCE loss (Oord et al., 2018)) to align an image with its corresponding textual description in the feature space. VLPs aim to improve multimodal task performance by pre-training on large-scale image-to-text pairs (Li et al., 2022). Several recent methods utilize pre-trained object detectors with

region features as a foundation for obtaining vision-language representations (Chen et al., 2020b). There are two primary types of VLPs depending on their architectures: fused and aligned (Zhang et al., 2022a). Fused VLPs, such as ALBEF and TCL (Yang et al., 2022), utilize distinct unimodal encoders to handle token embeddings and visual characteristics separately. They subsequently employ a multimodal encoder to produce integrated multimodal embeddings by combining image and text embeddings. Conversely, aligned VLPs such as CLIP are composed solely of unimodal encoders that have separate embeddings for image and text modalities. This research specifically examines widely used architectures, including both fused and aligned.

**Adversarial Attacks and Robustness in VLPs.** Adversarial attacks aim to deceive deep learning models into misclassifying an input (Szegedy et al., 2013). While previous work is centered around image classification, recent studies show that VLPs are also vulnerable to adversarial attacks. For example, Xu et al. (2018) investigated attacks on visual question-answering models by altering the image modality. Agrawal et al. (2018), and Shah et al. (2019) focused on disrupting vision-language models through text modality perturbations. Zhang et al. (2022a) offered key insights into the development of multimodal attacks and improving model robustness by exploring VLPs. Building on this, Lu et al. (2023); He et al. (2023), and Han et al. (2023) worked on enhancing the transferability of multimodal AEs by leveraging cross-modal interactions, data augmentation, and optimal transport theory. Furthermore, Yin et al. (2023b), and Zhou et al. (2023) build on Zhang et al. (2022a) by crafting modality-aligned perturbations that improve transferability between downstream tasks. Despite these advances, many attack techniques remain specialized for classification tasks and may not generalize well to retrieval, captioning, or grounding. Therefore, we adopted the adversarial attacks presented in Zhang et al. (2022a) as our baseline.

Recent efforts have explored enhancing the adversarial robustness of vision-language models through prompt tuning and training strategies. Zhang et al. (2024b) and Li et al. (2024) improve CLIP's resilience by learning robust textual prompts aligned with adversarial image embeddings. Zhou et al. (2024) introduces adversarial text supervision to balance cross-modal alignment and uni-modal discrimination. Wang et al. (2024a) adds an auxiliary branch to align adversarial outputs between the target and pre-trained models, reducing overfitting in zero-shot settings. Wang et al. (2024b) adopts a two-phase adversarial training regime, starting with lightweight pre-training, followed by high-resolution fine-tuning. However, the high computational overhead of these methods poses challenges for scaling to large models and datasets.

**Geometric Methods and Geometric Adversarial Detection in Unimodal Models.** $k$-NN (Cover & Hart, 1967) is a nonparametric algorithm that classifies points based on the majority label of their nearest neighbors, offering a simple yet powerful method for pattern recognition and regression. LID models the intrinsic dimensionality (Karger & Ruhl, 2002; Houle et al., 2012; Houle, 2013; 2017; Amsaleg et al., 2015) near a point by analyzing the growth rate of nearby data, providing insights into local geometric structures within a dataset. Mahalanobis distance (McLachlan, 1999) incorporates data covariance to measure similarity, enabling scale-invariant and correlation-sensitive evaluations that are effective for identifying outliers or understanding feature relationships. KDE estimates the probability density function of data in a nonparametric manner, using kernel functions and adaptive bandwidths to achieve smooth and flexible density representations (Botev et al., 2010).

Several studies have employed geometric approaches to detect AEs in unimodal classification models. Grosse et al. (2017) introduced the Maximum Mean Discrepancy (MMD), a kernel-based statistical test that distinguishes AEs from a model's training data. As an alternative to KDE, Ma et al. (2018) employed LID to evaluate the distance distribution of an input relative to its neighbors, capturing the local complexity of the sample's surrounding space. Lee et al. (2018) proposed using the Mahalanobis distance, using Gaussian discriminant analysis to detect out-of-distribution and adversarial samples through a generative classifier, offering a more refined confidence score than the traditional softmax classifier. Cohen et al. (2020) further explored $k$-NN for adversarial detection. While these methods have shown promise in unimodal settings, their effectiveness in VLPs remains unexplored.

## 3 GEODETECT

In this section, we introduce GeoDetect, a geometric framework for detecting AEs by analyzing the properties of embedding geometry in VLPs. We first provide the formal problem definition in Section 3.1, which is followed by a theoretical analysis in Section 3.2.

## 3.1 GeoDetect Framework

**Problem Setup.** Let $\mathcal{D}_c = \{(x_i, t_i)\}_{i=1}^N$ be a clean dataset of $N$ i.i.d. image-text pairs, where $x_i$ and $t_i$ denote the clean image and text, respectively. The adversarial dataset consists of perturbed samples and is defined as $\mathcal{D}_a = \{(x_i', t_i)\}_{i=1}^N$ when only the image is perturbed, or $\mathcal{D}_a = \{(x_i', t_i')\}_{i=1}^N$ when both modalities are perturbed, with labels $y_i \in \{0, 1\}$ indicating benign ($y_i = 0$) or adversarial ($y_i = 1$). We define the embeddings as $z_I = E_I(x)$ for image, $z_T = E_T(t)$ for text, and in the case of fused VLPs, $z_M = E_M(z_I, z_T)$ as the multimodal representation. Clean embeddings are denoted as $Z_c = \{z_i\}_{i=1}^N$, where $z_i$ represents either $z_I$ or $z_M$.

Our goal is to accurately detect perturbed samples, particularly those in which the image, or both the image and text modalities, have been adversarially modified. Given a query $(x_i, t_i)$ and a reference batch $\{(x_j, t_j)\}_{j=1}^n$, with $n$ denoting the batch size, we evaluate the detection function:

$$f((x_i, t_i), (x_j, t_j)_{j=1}^n) = \mathcal{H}(\text{Metric}(z_i, \{z_j\}_{j=1}^n)), \tag{1}$$

where $\{z_j\}_{j=1}^{n, j\neq i}$ denotes a set of clean reference embeddings, $n$ is the batch size, and $\mathcal{H}$ represents the decision function. The function $\text{Metric}(\cdot, \cdot)$ represents a geometric measure, such as LID, $k$-NN, KDE, or Mahalanobis distance. In this paper, we adopt the maximum likelihood estimation (MLE) of LID (Amsaleg et al., 2015), and throughout the paper, we use "LID" to refer to this estimated quantity. The binary classification $f(\cdot, \cdot)$ then determines whether the input is adversarial or clean using computed score.

**GeoDetect Pipeline.** GeoDetect comprises three primary steps: generation, extraction, and detection. Following prior work (Ma et al., 2018), we assume that the defender has access to a subset of the data, and that the initial dataset $\mathcal{D}_c$ is free of AEs.

In the first step of the process, generation, AEs are created from clean samples using different adversarial attacks. Given the clean dataset, we generate perturbed image–text pairs ($x_i'$ and $t_i'$), resulting in a balanced set with equal proportions of clean and adversarial samples. A detailed description of adversarial-sample generation, including the formulas and settings for each attack type, is provided in Appendix A.1.

In the extraction step, we begin by extracting clean image embeddings, $z_I$, and multimodal embeddings, $z_M$, as well as the corresponding adversarial embeddings $z_I'$ and $z_M'$. To ensure scalability on large datasets, we use minibatch sampling to estimate local geometric properties, following the approach in Ma et al. (2018), which has been shown to provide reliable approximations of neighborhood statistics. To compute geometric scores of a target embedding $z_i$, we randomly sample a batch of clean embeddings $\{z_j\}_{j=1}^n$ as reference points for the computation of different metrics. These reference points can be either $z_I$ or $z_M$, depending on whether the detection is performed in the image or multimodal space. Using this reference batch, scores will be computed for both clean $z_i$ and adversarial embeddings $z_i'$ using the following metrics:

$$Metric(\cdot, \cdot) = \begin{cases} \text{k-NN}\big(z_i, \{z_j\}_{j=1}^n\big) = \dfrac{1}{k}\sum_{j=1}^k r_j(z_i), \quad r_j(z_i) = \|z_i - z_j\|_2, \\[2ex] \widehat{\text{LID}}\big(z_i, \{z_j\}_{j=1}^n\big) = \Big(-\dfrac{1}{k}\sum_{j=1}^k \log \dfrac{r_j(z_i)}{r_{\max}(z_i)}\Big)^{-1}, \\[2ex] \text{KDE}\big(z_i, \{z_j\}_{j=1}^n; H\big) = \dfrac{1}{n}\sum_{j=1}^n K_H(z_i, z_j), \\[2ex] \text{Mahal}\big(z_i, \{z_j\}_{j=1}^n\big) = \sqrt{(z_i - \mu)^\mathsf{T}\Sigma^{-1}(z_i - \mu)}. \end{cases} \tag{2}$$

For the Mahalanobis distance, the mean vector $\mu$ and covariance matrix $\Sigma$ are computed using the clean dataset $\mathcal{D}_c$. Similarly, for KDE, the kernel function $K_H(\cdot, \cdot)$ is estimated based on the same clean data. For all metrics, the embedding-level scores are computed as $s_i = \text{Metric}(z_i, \{z_j\}_{j=1}^n)$ for clean samples, and $s_i' = \text{Metric}(z_i', \{z_j\}_{j=1}^n)$ for adversarial samples, where the reference points $\{z_j\}_{j=1}^n$ are clean embeddings samples. For LID, we follow a layer-wise extraction strategy as proposed by Ma et al. (2018). In fused VLPs, we additionally compute the LID of the multimodal encoder $z_M$ as an additional feature alongside the image encoder layers, improving detection performance against multimodal attacks. The complete procedure for computing these values for adversarial

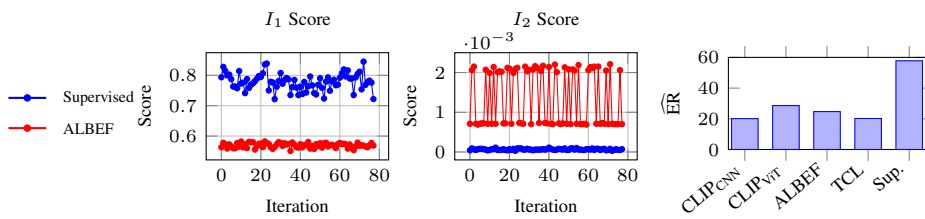

(a) Isotropy scores ($I_1$, $I_2$) across iterations.     (b) Normalized ER comparison.

Figure 2: Comparison of isotropy metrics and effective ranking for VLPs vs. supervised learning. Iteration refers to the batch index during evaluation.

image detection is outlined in Algorithm 1 in Appendix A.2. After the extraction step, the extracted clean and adversarial scores are denoted as $s_i \in S_{(N,l)}$ and $s'_i \in S'_{(N,l)}$, where $l = 1$ for $k$-NN, Mahalanobis, and KDE, and $l$ represents the number of layers for LID. These extracted scores serve as input features for the subsequent detection phase.

We frame adversarial example detection via a decision function $\mathcal{H}$. We split the extracted scores into a calibration subset (to set thresholds or train a classifier) and a test subset. For $k$-NN, Mahalanobis, and KDE, $\mathcal{H}$ is a threshold rule on the scalar score $s_i = \text{Metric}\big(z_i, \{z_j\}_{j=1}^n\big)$: $\mathcal{H}(s_i) = \mathbb{I}(s_i > \tau)$, where $\tau$ is chosen on the training split. For LID, $\mathcal{H}$ is a logistic regression trained on multi-layer LID features (Appendix A.2). At test time, given $(x_i, t_i)$, we extract embeddings $(z_I, z_T)$, compute the chosen metric with respect to a clean reference batch $\{z_j\}_{j=1}^n$, and apply $\mathcal{H}$ (threshold or logistic) to decide if the input is adversarial. The efficiency of GeoDetect is reported in Appendix A.4.

## 3.2 GEODETECT THEORETICAL ANALYSIS

In this subsection, we develop GeoDetect's theoretical foundations by formalizing core assumptions and deriving its key technical results. We begin by empirically verifying that VLP embeddings are anisotropic, concentrated along a few dominant directions, which motivates our assumptions. Building on this, we show that the principal directions of adversarial embeddings differ significantly from those of clean embeddings, effectively pushing them off the manifold. Our theoretical analysis explains why geometric scores are particularly well-suited for adversarial detection in VLPs: they quantify off-manifold deviations that adversarial perturbations inherently induce.

### 3.2.1 ANISOTROPIC EMBEDDING SPACE

Understanding the geometric structure of the embedding space is crucial for identifying the fundamental differences between clean and adversarial embeddings in VLPs. Liang et al. (2022) showed that CLIP features lie within a low-aperture cone, concentrating in a narrow angular region of high-dimensional space. Building on this, Levi & Gilboa (2024) found that CLIP's embedding space forms a double-ellipsoid geometry, with image and text embeddings located on distinct ellipsoidal shells.

Consequently, we expect that other VLPs also exhibit anisotropic embedding spaces, consistent with the patterns observed in CLIP, and to formally quantify this, we adopt two established isotropy measures, $I_1$ and $I_2$ (Wang et al., 2019). Let $Z \in R^{N \times D}$ be the matrix of embedding vectors. The measures are defined as:

$$I_1(Z) = \frac{\min_{v \in V} P(v)}{\max_{v \in V} P(v)}, \quad I_2(Z) = \sqrt{\frac{\sum_{v \in V}\big(P(v) - \bar{P}(v)\big)^2}{|V|\bar{P}(v)^2}}, \quad (3)$$

where $V$ is the set of eigenvectors of $Z^T Z$, and $P : \mathbb{R}^D \to \mathbb{R}^+$ is the partition function $P(v) = \sum_{i=1}^n \exp(\langle v, z_i \rangle)$ (Mu & Viswanath, 2018). For an embedding matrix $Z$ to be isotropic, $P(v)$ should be approximately constant for any unit vector $v$ (Arora et al., 2016). Then, the second measure, $I_2(Z)$, is the normalized standard deviation of the partition function $P(v)$, where $\bar{P}(v)$ is the average value of $P(v)$. In this formulation, $I_1(Z)$ is bounded between 0 and 1 ($I_1(Z) \in [0, 1]$), while $I_2(Z)$ is non-negative ($I_2(Z) \geq 0$); lower $I_1(Z)$ and higher $I_2(Z)$ indicate stronger anisotropy.

To investigate the embedding space of the VLPs (specifically ALBEF here, with other VLPs discussed in Appendix B.1), we empirically evaluate $I_1(Z)$ and $I_2(Z)$. As a reference, we include a supervised classifier (ResNet-50) trained on ImageNet. We compute $I_1(Z)$ and $I_2(Z)$ across iterations (x-axis) and report the corresponding metrics on the y-axis in Figure 2a. The results show that $I_1(Z)$ is lower and $I_2(Z)$ is higher for VLPs, shown here for ALBEF, compared to the supervised classifier, indicating stronger anisotropy in VLPs embedding space. Consistent trends for other VLPs are reported in Appendix B.1, supporting the assumptions underlying our theoretical analysis.

Another metric for evaluation of isotropy is effective rank (ER), a spectral measure of dimensionality that reflects how singular values are distributed, providing a more complex perspective compared to the traditional rank (Roy & Vetterli, 2007). Mathematically, the ER of a matrix $Z$ is defined based on the spectral entropy of its normalized singular values. Let $\sigma_i$ be the singular values of $Z$, and $\hat{\sigma}_i$ represent the normalized singular values $\hat{\sigma}_i = \dfrac{\sigma_i}{\sum_j \sigma_j}$, where $\sum_i \hat{\sigma}_i = 1$. The spectral entropy and the normalized ER, denoted as $\widehat{\text{ER}}$ (scaled by the logarithm of the dimension $D$), are given by:

$$H = -\sum_i \hat{\sigma}_i \log \hat{\sigma}_i \Rightarrow \widehat{\text{ER}}(Z) = \frac{\exp(H)}{\log D}. \tag{4}$$

In the isotropic setting, where all singular values are equal, the estimated effective rank $\widehat{\text{ER}}$ approaches the full rank of the matrix. In contrast, when the singular values are concentrated in a few dimensions, $\widehat{\text{ER}}$ is substantially lower, indicating anisotropy. Figure 2b compares the normalized effective rank of various VLPs with that of a supervised ResNet-50 trained on ImageNet V2. The results show that VLPs consistently exhibit lower $\widehat{\text{ER}}$, confirming that their embedding spaces are more anisotropic than those of supervised models.

### 3.2.2 ADVERSARIAL PERTURBATION EFFECT ON GEOMETRY

In this subsection, we establish the theoretical assumptions and provide key analyses underpinning our geometric-based detection methodology. Let $\Sigma \in R^{D \times D}$ denote the covariance matrix of the clean embedding distribution. We denote clean embeddings as $z_i \in Z_c$, and adversarial embeddings as $z_i' \in Z_a$. We further represent the distributions of clean and adversarial embeddings by $p(z)$ and $q(z')$, respectively. To characterize the geometric deviation induced by adversarial perturbations, we define the distance of a target embedding (clean or adversarial) to a randomly sampled clean embedding $z_u$, where $u \neq i$, as:

$$Dist_c = \|z_i - z_u\| \quad \text{and} \quad Dist_a = \|z_i' - z_u\|.$$

We analyze the expected distances $\mathbb{E}[Dist_c]$ and $\mathbb{E}[Dist_a]$, and formally demonstrate that adversarial embeddings have higher expected distances, an observation underlying the effectiveness of geometric metrics in adversarial detection. We first state two assumptions that ground our theoretical analysis:

**Assumption 3.1** (Anisotropic Covariance). The covariance matrix $\Sigma \in \mathbb{R}^{D \times D}$ of the clean embedding space is positive-definite and anisotropic, specifically $\Sigma \neq cI$ for any scalar constant $c$, indicating anisotropy property. Consequently, its eigenvalues vary significantly across dimensions $(\sigma_1 \gg \sigma_2 \gg ... \gg \sigma_D)$.

**Assumption 3.2** (Manifold Proximity). Clean embeddings reside on a manifold $\mathcal{M}$, such that the distance of a data point $z_i$ from the manifold satisfies, satisfying a proximity condition $\|z_i - \mathcal{M}\| \leq \alpha$. Additionally, given the high dimensionality of the embeddings, we assume Gaussian distributions for both clean and adversarial embeddings: $p(z) \sim \mathcal{N}(\mu_z, \Sigma), \quad q(z') \sim \mathcal{N}(\mu_{z'}, \Sigma')$.

Assumption 3.1, verified in Section 3.2.1, explains the anisotropic geometry observed in VLP embeddings. Assumption 3.2, built on the manifold hypothesis (Bengio et al., 2013), distinguishes clean embeddings that lie near the manifold from adversarial embeddings that deviate from it. To justify Assumption 3.2, we note that although data may be globally non-Gaussian or manifold-valued, it is standard to analyze them through local neighborhoods: non-Gaussian structures often appear approximately Gaussian when viewed locally, and curved manifolds can be locally approximated by Euclidean spaces (Lee, 2006). As emphasized by Zhao et al. (2007), even globally non-Gaussian or manifold-valued data exhibit locally Gaussian behavior, since any curved manifold is locally Euclidean. Building on these assumptions, and inspired by Theorem 1 in Zhang et al. (2024a), we derive the optimal adversarial embedding.

**Lemma 3.3.** *Following Assumption 3.2, let clean and adversarial embeddings follow $p(z) \sim \mathcal{N}(\mu_z, \Sigma)$ and $q(z') \sim \mathcal{N}(\mu_{z'}, \Sigma')$, respectively, where $\Sigma$ is a fixed positive-definite covariance. Maximizing the KL divergence $\mathrm{KL}(q\|p)$ is approximately equivalent to maximizing the quadratic form of $(z_i' - z_i)^\top \Sigma^{-1} (z_i' - z_i)$, which can be transformed into a Lagrangian minimization optimization problem, which has an optimal closed-form solution:*

$$z_i'^* = (\Sigma + \lambda I)^{-1} \lambda z_i, \quad \lambda > 0, \tag{5}$$

*where $\lambda$ is the Lagrange multiplier.*

The proof of Lemma 3.3 appears in Appendix B.2.

**Lemma 3.4.** *Following Assumption 3.1 and Lemma 3.3, let $\mathcal{M}$ represent the data manifold formed by clean embeddings within a local batch. When the data is perturbed, and assuming that clean embeddings lie close to the manifold $\mathcal{M}$, the resulting embeddings deviate from the manifold $\mathcal{M}$, thus characterized as off-manifold. Specifically, given the optimal adversarial embedding $z_i'^* = (\Sigma + \lambda I)^{-1} \lambda z_i, \quad \lambda > 0$, the deviation from the manifold satisfies $\|z_i' - \mathcal{M}\| \geq \gamma$, where $\gamma > \alpha \geq 0$ defines the minimum separation threshold for off-manifold data.*

Lemma 3.4 shows that adversarial embeddings leave the clean manifold by suppressing tangent components and amplifying normal components (proof in Appendix B.3). We verify this with four diagnostics in Appendix B.6: reconstruction error using top-$K$ principal components fitted on clean data, Singular Value Decomposition (SVD) tail energy, KL divergence along principal components, and lower rank correlation between clean and adversarial representations; all confirm off-manifold deviation.

**Theorem 3.5** (Expected Distance Gap). *Following Lemmas 3.3 and 3.4, let $z_i \sim p(\cdot)$ be a clean embedding satisfying $\|z_i - \mathcal{M}\| \leq \alpha$, and let $z_i'$ be an adversarial embedding satisfying $\|z_i' - \mathcal{M}\| \geq \gamma$ with $\gamma > 3\alpha$. Then*

$$\mathbb{E}_{z_u \sim p(\cdot)}\big[\|z_i' - z_u\|\big] > \mathbb{E}_{z_u \sim p(\cdot)}\big[\|z_i - z_u\|\big]. \tag{6}$$

Theorem 3.5 implies that given a query embedding, a large distance to randomly sampled clean embeddings strongly indicates adversarial perturbation. This justifies the effectiveness of geometric-based metrics for adversarial detection. Proof of Theorem 3.5 is included in Appendix B.4, and details on the connection between Theorem 3.5 and Lemma 3.4 to different geometric-based approaches are provided in Appendix B.5. Specifically, we demonstrate that under mild assumptions, AEs are expected to exhibit higher LID, $k$-NN, and Mahalanobis scores, along with lower KDE scores. Empirical evidence supporting Lemma 3.4 and Theorem 3.5 is provided in Appendix B.7.

## 4 EXPERIMENTS

We evaluate GeoDetect on standard VLP tasks, including zero-shot classification and image-text retrieval. We compare against MCM (Ming et al., 2022), which is designed for CLIP-based classification and detects out-of-distribution inputs via softmax-normalized similarity scores. To the best of our knowledge, this is the only existing score-based zero-shot baseline applicable to AE detection in multimodal models. While relevant for classification-based VLP settings, its reliance on discrete class labels makes it incompatible with retrieval tasks.

**Datasets and Models.** We evaluate zero-shot classification with ImageNet (Deng et al., 2009), CIFAR10, CIFAR100 (Krizhevsky et al., 2009), STL-10 (Coates et al., 2011), and Food-101 (Bossard et al., 2014), as the standard datasets for zero-shot classification. Following Radford et al. (2021), we use class prompts of the form "a photo of a $c$", where $c$ is the name of the class. For image-text retrieval, we conduct experiments on commonly used datasets, Flickr30K (Young et al., 2014) and MS-COCO (Lin et al., 2014). We consider two types of VLPs: aligned and fused. For aligned VLPs, we evaluate $\mathrm{CLIP_{ViT}}$ (using ViT-B/16) and $\mathrm{CLIP_{CNN}}$ (using ResNet-50) (Radford et al., 2021). For fused VLPs, we examine ALBEF (Li et al., 2021) and TCL (Yang et al., 2022), which consist of separate image, text, and multimodal encoders.

**Metric and Adversarial Attack.** We assess performance using two standard metrics: (1) the false positive rate at 95% true positive rate (FPR95), and (2) the area under the receiver operating characteristic curve (AUC). We follow Sep-Attack and Co-Attack methods (Zhang et al., 2022a)

Table 1: Results on zero-shot classification performance using the area under the receiver operating characteristic curve (AUC) and the false positive rate at 95% true positive rate (FPR95). Higher AUC ($\uparrow$) and lower FPR95 ($\downarrow$) values indicate more accurate detection.

| Model | Method | Attack | CIFAR10 | | CIFAR100 | | ImageNet1k | | STL10 | | Food101 | |
|---|---|---|---|---|---|---|---|---|---|---|---|---|
| | | | AUC | FPR95 | AUC | FPR95 | AUC | FPR95 | AUC | FPR95 | AUC | FPR95 |
| CLIP$_{CNN}$ | MCM | Sep$_{uni}$ | 65.47 | 82.88 | 41.13 | 94.15 | 86.10 | 60.35 | 95.82 | 17.92 | 91.70 | 40.18 |
| | | Co-Attack | 67.10 | 79.54 | 43.99 | 93.21 | 80.83 | 68.38 | 94.10 | 25.64 | 82.14 | 64.38 |
| | GeoDet-LID | Sep$_{uni}$ | 100 | 0.00 | 100 | 0.00 | 99.31 | 1.87 | 100 | 0.00 | 99.98 | 0.06 |
| | | Co-Attack | 100 | 0.00 | 100 | 0.00 | 99.50 | 1.62 | 100 | 0.00 | 99.95 | 0.08 |
| | GeoDet-$k$-NN | Sep$_{uni}$ | 100 | 0.00 | 100 | 0.00 | 99.65 | 1.62 | 100 | 0.00 | 100 | 0.00 |
| | | Co-Attack | 100 | 0.00 | 100 | 0.00 | 99.67 | 0.89 | 100 | 0.00 | 100 | 0.00 |
| | GeoDet-Mah. | Sep$_{uni}$ | 100 | 0.00 | 100 | 0.00 | 96.62 | 9.32 | 99.88 | 0.33 | 99.79 | 1.16 |
| | | Co-Attack | 100 | 0.00 | 100 | 0.00 | 97.28 | 7.97 | 99.80 | 0.59 | 99.38 | 2.32 |
| | GeoDet-KDE | Sep$_{uni}$ | 100 | 0.00 | 100 | 0.00 | 98.72 | 7.24 | 99.87 | 0.26 | 100 | 0.00 |
| | | Co-Attack | 100 | 0.00 | 100 | 0.00 | 99.33 | 2.81 | 99.85 | 0.33 | 100 | 0.00 |
| ALBEF | MCM | Sep$_{uni}$ | 91.20 | 29.02 | 82.80 | 49.19 | 92.15 | 25.38 | 96.83 | 16.02 | 90.26 | 37.03 |
| | | Sep$_{multi}$ | 47.43 | 98.23 | 33.55 | 99.56 | 63.03 | 97.59 | 65.32 | 86.60 | 41.98 | 99.46 |
| | | Co-Attack | 93.34 | 24.64 | 82.67 | 47.27 | 92.14 | 24.94 | 96.45 | 19.70 | 81.29 | 74.70 |
| | GeoDet-LID | Sep$_{uni}$ | 100 | 0.00 | 99.97 | 0.05 | 91.85 | 29.41 | 99.64 | 1.62 | 99.87 | 0.67 |
| | | Sep$_{multi}$ | 99.96 | 0.20 | 99.85 | 0.44 | 78.77 | 67.68 | 96.63 | 15.65 | 92.31 | 33.27 |
| | | Co-Attack | 100 | 0.00 | 99.98 | 0.05 | 93.85 | 20.07 | 99.85 | 0.69 | 99.92 | 0.42 |
| | GeoDet-$k$-NN | Sep$_{uni}$ | 100 | 0.00 | 100 | 0.00 | 98.60 | 7.23 | 99.97 | 0.19 | 99.98 | 0.04 |
| | | Sep$_{multi}$ | 99.27 | 3.05 | 99.21 | 3.25 | 51.92 | 93.61 | 75.95 | 75.75 | 86.46 | 50.96 |
| | | Co-Attack | 100 | 0.00 | 100 | 0.00 | 98.64 | 7.33 | 99.96 | 0.19 | 99.98 | 0.04 |
| | GeoDet-Mah. | Sep$_{uni}$ | 100 | 0.00 | 100 | 0.00 | 99.94 | 0.20 | 100 | 0.00 | 100 | 0.00 |
| | | Sep$_{multi}$ | 100 | 0.00 | 100 | 0.00 | 81.41 | 64.82 | 99.25 | 3.19 | 99.16 | 3.92 |
| | | Co-Attack | 100 | 0.00 | 100 | 0.00 | 99.93 | 0.25 | 100 | 0.00 | 100 | 0.00 |
| | GeoDet-KDE | Sep$_{uni}$ | 99.38 | 0.71 | 100 | 0.00 | 96.78 | 16.93 | 99.70 | 1.06 | 99.95 | 0.16 |
| | | Sep$_{multi}$ | 99.24 | 0.86 | 99.85 | 0.81 | 66.83 | 81.75 | 88.63 | 66.19 | 87.61 | 49.27 |
| | | Co-Attack | 99.38 | 0.76 | 100 | 0.00 | 96.67 | 18.09 | 99.72 | 1.00 | 99.94 | 0.16 |

due to their applicability to different models and tasks. [CLS] is a class embedding widely used in pre-trained models for downstream tasks, which is the target of the attack in our evaluation. Sep-Attack perturbs each modality independently, while Co-Attack jointly targets both modalities. For image-focused attacks, we evaluate two variants of Sep-Attack: Sep$_{uni}$, which targets unimodal embeddings, and Sep$_{multi}$, which targets the fused multimodal representation (applicable only to fused VLPs). For image attacks, consistent with (Zhang et al., 2022a), we adopt PGD-style perturbations constrained in the $\ell_\infty$ norm, and a BERT-style (Li et al., 2020) attack strategy for text attack. The maximum perturbation $\epsilon_i$ is set to $8/255$, with the step size of $1.25$, and 10 iterations. For text, the perturbation budget is set to 1 token. Detailed attack configurations and success rates are reported in Appendix A.1 and Appendix A.5, respectively.

**Settings and Layers.** For CLIP$_{CNN}$ and CLIP$_{ViT}$, we use batch size 128 with $k=100$ for LID, $k=10$ for $k$-NN, and a Gaussian KDE bandwidth of $0.1$. For ALBEF and TCL, batch size is 64 with $k=40$ for LID, $k=10$ for $k$-NN, and the same KDE bandwidth. Adversarial examples are generated from the entire test set. The resulting mixed dataset (clean and adversarial) is then randomly split into 80% for calibration (threshold fitting / LID training) and 20% for evaluation. Detailed layer selection is provided in Appendix A.3, with layer sensitivity in Appendix C.3.

## 4.1 EXPERIMENTAL RESULTS

**Performance of GeoDetect in Zero-Shot Classification.** As shown in Table 1, our geometric approaches consistently outperform the MCM method across all datasets in CLIP$_{CNN}$, achieving lower FPR and higher AUC. This highlights GeoDetect's effectiveness for AE detection. Among the evaluated metrics, $k$-NN surpasses other metrics (particularly Mahalanobis and KDE) in CLIP$_{CNN}$, with LID showing comparable performance to $k$-NN in this context. On ALBEF, Mahalanobis slightly outperforms other metrics, particularly KDE, highlighting its sensitivity to image-level

Table 2: Results on image-text retrieval with Flickr30k and COCO dataset evaluated using the area under the receiver operating characteristic curve (AUC) and the false positive rate at 95% true positive rate (FPR95). Higher AUC (↑) and lower FPR95 (↓) values indicate more accurate detection.

(a) Results for $\text{CLIP}_{\text{CNN}}$ and ALBEF Models

| Model | Method | Attack | Dataset | | | |
| | | | Flickr30k | | COCO | |
| | | | AUC | FPR95 | AUC | FPR95 |
| $\text{CLIP}_{\text{CNN}}$ | LID | $\text{Sep}_{\text{uni}}$ | 98.45 | 4.52 | 99.54 | 1.46 |
| | | Co-Attack | 98.90 | 4.52 | 99.50 | 1.56 |
| | $k$-NN | $\text{Sep}_{\text{uni}}$ | 99.99 | 0.00 | 99.97 | 0.00 |
| | | Co-Attack | 99.97 | 0.00 | 99.95 | 0.02 |
| ALBEF | LID | $\text{Sep}_{\text{uni}}$ | 94.99 | 23.83 | 91.80 | 35.88 |
| | | $\text{Sep}_{\text{multi}}$ | 74.26 | 78.75 | 79.85 | 64.51 |
| | | Co-Attack | 93.80 | 27.98 | 91.49 | 35.58 |
| | $k$-NN | $\text{Sep}_{\text{uni}}$ | 99.75 | 1.05 | 98.54 | 5.67 |
| | | $\text{Sep}_{\text{multi}}$ | 54.84 | 92.46 | 57.02 | 88.27 |
| | | Co-Attack | 99.88 | 0.50 | 98.73 | 6.84 |

(b) Results for $\text{CLIP}_{\text{ViT}}$ and TCL Models

| Model | Method | Attack | Dataset | | | |
| | | | Flickr30k | | COCO | |
| | | | AUC | FPR95 | AUC | FPR95 |
| $\text{CLIP}_{\text{ViT}}$ | LID | $\text{Sep}_{\text{uni}}$ | 99.37 | 1.51 | 99.98 | 0.20 |
| | | Co-Attack | 96.55 | 25.63 | 99.06 | 5.08 |
| | $k$-NN | $\text{Sep}_{\text{uni}}$ | 100 | 0.00 | 100 | 0.00 |
| | | Co-Attack | 99.59 | 0.50 | 99.51 | 1.27 |
| TCL | LID | $\text{Sep}_{\text{uni}}$ | 90.88 | 40.93 | 89.32 | 42.52 |
| | | $\text{Sep}_{\text{multi}}$ | 84.72 | 56.47 | 83.95 | 58.16 |
| | | Co-Attack | 90.76 | 37.31 | 88.25 | 43.79 |
| | $k$-NN | $\text{Sep}_{\text{uni}}$ | 96.10 | 19.60 | 98.01 | 11.24 |
| | | $\text{Sep}_{\text{multi}}$ | 32.89 | 95.48 | 33.89 | 95.41 |
| | | Co-Attack | 96.59 | 15.07 | 98.05 | 11.73 |

perturbations, while LID performs comparably in multimodal attacks, emphasizing the value of incorporating multimodal embeddings. Despite ALBEF's multimodal design, image perturbations still yield detectable shifts in the embedding space, which Mahalanobis effectively captures by modeling the covariance of clean image features. Extended results for $\text{CLIP}_{\text{ViT}}$ and TCL in Appendix E.1 exhibit consistent patterns with those in this subsection.

**Performance of GeoDetect in Image-Text Retrieval.** We also evaluate image-text retrieval to demonstrate that GeoDetect applies beyond classification, without labeled data. Due to the lack of labels, we only examine the LID and $k$-NN distance, as they do not require labels. As shown in Table 2, the performance of all models is comparable to their classification results. For both the COCO and Flickr30k datasets, each image is annotated with five captions. To maintain consistency, as Co-Attack requires a matching prompt to simultaneously attack both the image and the associated text, we use the first caption as the target text.

**Extended Evaluation and Ablation Study.** Appendix B.7 provides empirical verification of GeoDetect, including visualizations showing clear separability between clean and adversarial samples via geometric scores. Sensitivity analyses over neighborhood size, sample availability, batch size, layer choice, and multimodal layers are presented in Appendix C, demonstrating robustness to these variations. Appendix D evaluates generalization to diverse attack backbones, the SGA attack (Lu et al., 2023), and adaptive attacks designed to evade detection; GeoDetect remains robust. Appendix E reports extended evaluations on additional models (TCL, $\text{CLIP}_{\text{ViT}}$) and a comparison with PIP (Zhang et al., 2024c) (a VQA-specific adversarial detector), showing that GeoDetect maintains its performance across models and achieves superior results to PIP.

## 5 CONCLUSION

In this paper, we propose the first task-free, theoretically grounded framework for detecting AEs in VLPs. By leveraging the anisotropic structure of VLP embedding spaces, we show through theoretical analysis that adversarial perturbations push embeddings into off-manifold regions, leading to fundamental geometric differences between clean and perturbed samples. Building on this insight, we introduce GeoDetect, a lightweight, model-agnostic detection method that applies simple geometric metrics to image or joint representations. GeoDetect generalizes across multiple tasks and leading VLP architectures, and achieves strong detection performance against a range of state-of-the-art adversarial attacks. Notably, GeoDetect remains effective even under adaptive attack settings, where adversaries are aware of the detection strategy and attempt to bypass it. This robustness, combined with its independence from task-specific logits or labels, makes GeoDetect well-suited for both classification and retrieval scenarios.

## ETHICS STATEMENT

This work studies adversarial detection in vision–language models using publicly available datasets. No private or personally identifiable data is used, and the proposed method strengthens model robustness against adversarial manipulation, contributing to trustworthiness in multimodal AI. Our primary aim in this research is to support secure and ethical applications. All research was conducted in compliance with the ICLR Code of Ethics.

## REPRODUCIBILITY STATEMENT

All theoretical claims are accompanied by complete proofs in Appendix B. Experimental settings and implementation details, including hyperparameters, are described in Section 4, while attacks' details are provided in Appendix A.1. Finally, the algorithm for GeoDetect is explicitly presented in Appendix A.2.

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

## A  ATTACKS AND ALGORITHM DETAILS

### A.1  ATTACKS ON VLPS

In this part, we provide an extended explanation of the adversarial attacks discussed in Section 3.1, including technical formulations and implementation details for the Sep-Attack, Co-Attack, and SGA attack strategies. These attacks are defined based on their perturbation targets within vision–language models, expanding the conceptual and mathematical foundations provided in the main paper.

In aligned VLPs (e.g., CLIP), attacks are constrained to unimodal embeddings, since only image and text encoders are accessible. In contrast, fused VLPs (e.g., ALBEF and TCL) allow perturbations on both unimodal and multimodal embeddings. These can be further categorized based on whether the entire embedding is perturbed (denoted $\text{Uni}_{\text{Full}}$ or $\text{Multi}_{\text{Full}}$) or only the [CLS] token representation (denoted $\text{Uni}_{\text{CLS}}$ or $\text{Multi}_{\text{CLS}}$).

In our experiments, we focus on $\text{Uni}_{\text{CLS}}$ image attacks for CLIP, and both $\text{Uni}_{\text{CLS}}$ and $\text{Multi}_{\text{CLS}}$ attacks for ALBEF and TCL. The [CLS] embedding plays a critical role in transformer-based models, as it is commonly used for downstream classification and retrieval tasks. Therefore, investigating the impact of attacks on the [CLS] embedding in VLPs is important. Although CLIP does not explicitly define a [CLS] token—especially in its ResNet-based variant—we treat the final image embedding as functionally equivalent to [CLS] for consistency across models. For $\text{CLIP}_{\text{ViT}}$, we directly use the [CLS] token. For simplicity, we refer to attacks on unimodal and multimodal [CLS] embeddings as $\text{Sep}_{\text{uni}}$ and $\text{Sep}_{\text{multi}}$, respectively, and omit explicit [CLS] notation in the paper and experiments section.

We evaluate three adversarial attack strategies: Sep-Attack and Co-Attack, based on the framework by Zhang et al. (2022a), and the recently proposed Set Guided Attack (SGA) (Lu et al., 2023). Sep-Attack perturbs the image and text modalities independently by maximizing the KL divergence between clean and perturbed embeddings. For text, adversarial changes are constrained to a limited number of tokens using a BERT-style attack strategy. In contrast, Co-Attack jointly optimizes perturbations across both modalities, pushing the embedding away from its original representation. Co-Attack is applicable to both aligned and fused VLPs, with gradient-based perturbations computed separately for each modality. SGA attack extends traditional image-text alignment to a set-level framework. By employing data augmentation, SGA constructs diverse image sets and pairs them with multiple textual descriptions, leveraging cross-modal guidance to enhance AEs transferability.

**Sep-attack**  Sep-Attack (Zhang et al., 2022a) is designed to perturb image and text modalities separately. As VLPs are often applied to non-classification tasks that lack explicit labels, the attack replaces the standard cross-entropy objective with a KL divergence loss ($\mathcal{L}$) between the embedding-wise representation to produce an adversarial perturbation:

$$\delta_I = \epsilon_I . sign(\nabla_{x'} \mathcal{L}(E_I(x^{'}), E_I(x))). \tag{7}$$

For perturbing the text modality, the text perturbations are generated as follows:

$$\delta_T = \arg \max_t (\|E_T(t^{'}) - E_T(t)\|) - t. \tag{8}$$

Maximum perturbation $\epsilon_T$ is set to the number of perturbed tokens in each prompt based on the BERT attack (Li et al., 2020). For attacks targeting multimodal embedding, the unimodal encoder is replaced with the multimodal encoder, denoted as $E_M(\cdot, \cdot)$. This attack setup is only applicable to fused VLPs like ALBEF, which incorporate a multimodal encoder. The image-based attack is as follows:

$$\delta_I = \epsilon_I . sign(\nabla_{x'} \mathcal{L}(E_M(E_I(x'), E_T(t), E_M(E_I(x), E_T(t))))). \tag{9}$$

**Co-Attack**  In Sep-attack, perturbing the text and image modalities independently may lead to suboptimal adversarial effectiveness. To overcome this challenge, Co-Attack (Zhang et al., 2022a) was developed to jointly optimize perturbations across both modalities. It aims to shift the perturbed multimodal embedding away from the original embedding or to maximize the discrepancy between the perturbed image and text embeddings. Co-Attack is applicable to both fused VLPs and aligned VLPs, making it suitable for attacking both multimodal and unimodal embeddings. The unimodal

attack aims to find the perturbation $\delta_{\mathrm{I}}$ that satisfies:

$$\arg\max_{\delta_{\mathrm{I}}} \mathcal{L}(E_{\mathrm{I}}(x'), E_{\mathrm{T}}(t)) + \beta_1 \mathcal{L}(E_{\mathrm{I}}(x'), E_{\mathrm{T}}(t')). \tag{10}$$

The attack on multimodal embedding is as follows:

$$\arg\max_{\delta_{\mathrm{I}}} \mathcal{L}(E_{\mathrm{M}}(E_{\mathrm{I}}(x'), E_{\mathrm{T}}(t')), E_{\mathrm{M}}(E_{\mathrm{I}}(x), E_{\mathrm{T}}(t')))$$
$$+ \beta_2 \mathcal{L}(E_{\mathrm{M}}(E_{\mathrm{I}}(x'), E_{\mathrm{T}}(t')), E_{\mathrm{M}}(E_{\mathrm{I}}(x), E_{\mathrm{T}}(t))). \tag{11}$$

$\beta_1$ and $\beta_2$ are hyperparameters that control the contributions of the second term.

**Set-level Guidance Attack (SGA)**  SGA introduces a set-level adversarial guidance mechanism, where perturbations are crafted to reduce the overall separability between sets of clean and adversarial embeddings. The feasible perturbation spaces for adversarial optimization are denoted by $B[x, \epsilon_{\mathrm{I}}]$ for images and $B[t_i, \epsilon_{\mathrm{T}}]$ for text, where $\epsilon_{\mathrm{I}}$ denotes the maximal perturbation bound for the image, and $\epsilon_{\mathrm{T}}$ denotes the maximal number of changeable words in the caption.

First, it generates corresponding adversarial captions for all captions in the text set $t_i$, forming an adversarial caption set $t_i' = \{t_1', t_2', \ldots, t_M'\}$. The adversarial caption $t_j'$ is constrained to be dissimilar to the original image $x$ in the embedding space. Next, the adversarial image $x'$ is generated by solving:

$$t_j' = \underset{t_j' \in B[t_j, \epsilon_{\mathrm{T}}]}{\mathrm{argmax}} \left( -\frac{E_{\mathrm{T}}(t_j') \cdot E_{\mathrm{I}}(x)}{\|E_{\mathrm{T}}(t_j')\| \|E_{\mathrm{I}}(x)\|} \right), x' = \underset{x' \in B[x, \epsilon_{\mathrm{I}}]}{\mathrm{argmax}} \left( -\sum_{j=1}^{M} \frac{E_{\mathrm{T}}(t_j')}{\|E_{\mathrm{T}}(t_j')\|} \sum_{s_i \in S} \frac{E_{\mathrm{I}}(g(x', s_i))}{\|E_{\mathrm{I}}(g(x', s_i))\|} \right) \tag{12}$$

Here, $g(x', s_i)$ denotes the resizing function that takes the adversarial image $x'$ and the scale coefficient $s_i$ as inputs. All resized versions of $x'$ are encouraged to be far from all adversarial caption set $t_j'$ in the embedding space. Finally, the adversarial caption $t'$ is generated as follows:

$$t' = \underset{t' \in B[t, \epsilon_t]}{\mathrm{argmax}} \left( -\frac{E_{\mathrm{T}}(t') \cdot E_{\mathrm{I}}(x')}{\|E_{\mathrm{T}}(t')\| \|E_{\mathrm{I}}(x')\|} \right), \tag{13}$$

## A.2 GEODETECT ALGORITHM

The details of the GeoDetect are presented in Algorithm 1. Specifically, line 2 corresponds to the AEs generation step described in Section 3, while lines 4 to 13 implement the extraction steps. Then, the detection phase is outlined in lines 16–18. For LID-based detection, we use the extracted scores matrix $S_{(N,l)}$, where $N$ denotes the number of samples and $l$ the number of layers from which features are derived. The extracted scores are then partitioned into calibration and test subsets. The calibration set is used to fit either a logistic regression (for LID) or apply a threshold-based decision rule (for $k$-NN, Mahalanobis, and KDE) for adversarial detection.

## A.3 LAYER SELECTION

For all the models except LID, we use the final layer, but for LID, following (Ma et al., 2018), we use multiple layers of image encoder and the multimodal encoder final layer (in the case of multimodal attack). To investigate which layers contribute most to adversarial detection across architectures in GeoDetect-LID, we perform a consistent layer-wise sampling strategy for both ViT and ResNet-based CLIP models. In the ViT model, we select one early residual block, two mid-level blocks, one late-stage block, and the final embedding. This covers the full semantic depth of the Transformer, from localized attention to abstract class-level representation. In the ResNet-based model, we include early convolutional features, a final residual block, and the final attention-pooled feature. Across both models, we find that mid-to-late layers tend to be the most discriminative for adversarial detection, as they encode both spatial and semantic context before final pooling or projection. However, including early layers improves detection sensitivity by exposing low-level shifts caused by perturbations. We also conducted an experiment in Appendix C.3 to evaluate the effect of different layers.

---

**Algorithm 1** GeoDetect: Geometric Detection of AEs in VLPs

---

**Input**: Pre-trained model with image encoder $E_{\text{I}}(\cdot)$, text encoder $E_{\text{T}}(\cdot)$, and (if fused) multimodal encoder $E_{\text{M}}(\cdot, \cdot)$; clean dataset $D_c = \{(x_i, t_i)\}_{i=1}^N$; set of extraction layers $L = \{l_1, l_2, \ldots, l_f\}$; and a chosen geometric metric (e.g., LID, $k$-NN, Mahalanobis, KDE). All metrics use clean embeddings as reference points.

**Output**: A detection model.

1: **for** $i = 1$ **to** $N$ **do**
2:    Generate adversarial pair: $(x_i', t_i') \leftarrow \text{ATTACK}(x_i, t_i)$
3:    **for** each layer $l \in L$ **do**
4:       Extract image embeddings: $z_{(i,l)} \leftarrow E_{\text{I}}^{(l)}(x_i)$
5:       Extract adversarial embeddings: $z_{(i,l)}' \leftarrow E_{\text{I}}^{(l)}(x_i')$
6:       Compute clean score: $s_{(i,l)} \leftarrow \text{METRIC}(z_{(i,l)}, \{z_j\}_{j=1}^n)$
7:       Compute adversarial score: $s_{(i,l)}' \leftarrow \text{METRIC}(z_{(i,l)}', \{z_j\}_{j=1}^n)$
8:    **end for**
9:    **if** attack targets multimodal embedding and metric is LID **then**
10:      $s_{(i,l+1)} \leftarrow \text{METRIC}(E_{\text{M}}(E_{\text{I}}(x_i), E_{\text{T}}(t_i)), \{z_j\}_{j=1}^n)$
11:      $s_{(i,l+1)}' \leftarrow \text{METRIC}(E_{\text{M}}(E_{\text{I}}(x_i'), E_{\text{T}}(t_i')), \{z_j\}_{j=1}^n)$
12:    **end if**
13: **end for**
14: Concatenate clean/adversarial scores: $X \leftarrow [S, S']$
15: Create labels: $Y \leftarrow [\underbrace{0, \ldots, 0}_{N}, \underbrace{1, \ldots, 1}_{N}]$
16: Detection model on $(X, Y)$

---

## A.4 GEODETECT EFFICIENCY

On CLIP$_{\text{CNN}}$ with the CIFAR-10 dataset, feature extraction for detection methods such as MCM, KDE, Mahalanobis, and $k$-NN takes less than 2 minutes, while LID-based extraction requires approximately 9 minutes on an NVIDIA H100 GPU. A detailed breakdown of time costs is provided in Table 3. Compared to re-training or fine-tuning approaches for improving CLIP robustness, our framework is significantly more efficient. For reference, linear-probe CLIP (Radford et al., 2021) requires roughly 13 minutes, CoOp (Zhou et al., 2022) takes over 14 hours, and CLIP-Adapter (Peng et al., 2021) requires approximately 50 minutes, all reported on a single NVIDIA RTX 3090 GPU (Zhang et al., 2022b).

Table 3: Comparison of computation time (seconds) for different methods of the GeoDetect framework on CIFAR-10 using CLIP$_{\text{CNN}}$ and a single NVIDIA H100 GPU.

| Method | LID | $k$-NN | Mahalanobis | KDE | MCM |
|---|---|---|---|---|---|
| **Score** | 546.11 | 103.19 | 33.08 | 69.82 | 98.48 |

## A.5 EVALUATION OF ATTACK SUCCESS RATES

We evaluate attack success rates (ASR) on image retrieval (four models, two datasets) and zero-shot classification (two models, five datasets). Table 4 shows that attacks are highly effective in the retrieval setting, consistently achieving large ASR values. This indicates that VLPs are vulnerable to adversarial attacks in retrieval tasks. Table 5 reports results for zero-shot classification, again showing consistently high ASR across models and datasets.

Table 4: ASR results (ASR@1 and ASR@5) for the Image-Retrieval Task with Flickr30k and COCO test data in aligned VLPs ($\text{CLIP}_{\text{CNN}}$, $\text{CLIP}_{\text{ViT}}$) and fused VLPs (ALBEF, TCL).

| Model | Attack | Flickr30k | | COCO | |
|---|---|---|---|---|---|
| | | ASR@1 | ASR@5 | ASR@1 | ASR@5 |
| $\text{CLIP}_{\text{CNN}}$ | $\text{Sep}_{\text{uni}}$ | 96.58 | 93.76 | 97.32 | 95.67 |
| | Co-Attack | 97.60 | 95.21 | 98.75 | 97.44 |
| $\text{CLIP}_{\text{ViT}}$ | $\text{Sep}_{\text{uni}}$ | 93.85 | 86.94 | 97.11 | 95.21 |
| | Co-Attack | 94.97 | 90.12 | 98.59 | 97.06 |
| ALBEF | $\text{Sep}_{\text{uni}}$ | 98.50 | 96.45 | 99.45 | 98.62 |
| | $\text{Sep}_{\text{multi}}$ | 96.56 | 95.49 | 95.84 | 95.70 |
| | Co-Attack | 98.71 | 97.07 | 99.52 | 98.92 |
| TCL | $\text{Sep}_{\text{uni}}$ | 99.76 | 99.01 | 99.84 | 99.56 |
| | $\text{Sep}_{\text{multi}}$ | 70.07 | 68.41 | 67.06 | 68.66 |
| | Co-Attack | 99.24 | 98.55 | 99.79 | 99.51 |

Table 5: ASR results for the zero-shot classification task in $\text{CLIP}_{\text{CNN}}$ and $\text{CLIP}_{\text{ViT}}$.

| Dataset | $\text{CLIP}_{\text{CNN}}$ | | $\text{CLIP}_{\text{ViT}}$ | |
|---|---|---|---|---|
| | Sep-Attack | Co-Attack | Sep-Attack | Co-Attack |
| CIFAR10 | 80.11 | 98.43 | 78.93 | 99.71 |
| CIFAR100 | 90.33 | 100 | 92.77 | 100 |
| STL10 | 90.04 | 99.66 | 76.51 | 99.95 |
| ImageNet1k | 98.69 | 99.96 | 98.88 | 99.98 |
| Food101 | 99.16 | 100 | 99.80 | 100 |

# B    THEORETICAL ANALYSIS AND PROOFS

## B.1    ISOTROPY ANALYSIS OF DIFFERENT VLPs

In this section, we evaluate the $I_1$ and $I_2$ metrics introduced in Section 3.2.1 for the $\text{CLIP}_{\text{ViT}}$, ALBEF, and TCL, and compare them to supervised training. As shown in Figure 3, all three models exhibit lower $I_1$ values and higher $I_2$ values compared to the supervised mode, consistent with our expectations. These observations indicate that VLPs are less isotropic, or more anisotropic, than models trained with supervised learning.

## B.2    PROOF OF LEMMA 3.3

*Proof.* The KL divergence between two multivariate Gaussian distributions $\mathcal{N}(\mu', \Sigma')$ and $\mathcal{N}(\mu, \Sigma)$ is given by:

$$\text{KL}\big(\mathcal{N}(\mu', \Sigma') \| \mathcal{N}(\mu, \Sigma)\big) = \tfrac{1}{2}\Big[\text{tr}(\Sigma^{-1}\Sigma') + (\mu'-\mu)^\top \Sigma^{-1}(\mu'-\mu) - D + \ln\big(\frac{\det(\Sigma)}{\det(\Sigma')}\big)\Big], \quad (14)$$

where $D$ is the dimensionality of the space. Since $\Sigma$ is fixed, maximizing KL divergence over $z_i'$ reduces to maximizing the second term:

$$(z_i' - z_i)^\top \Sigma^{-1}(z_i' - z_i). \quad (15)$$

which is a positive-definite quadratic form in $(z_i' - z_i)$ and grows unbounded as $\|z_i'\| \to \infty$. To ensure a finite optimum, we introduce a constraint:

$$(z_i' - z_i)^\top \Sigma^{-1}(z_i' - z_i) \leq 1. \quad (16)$$

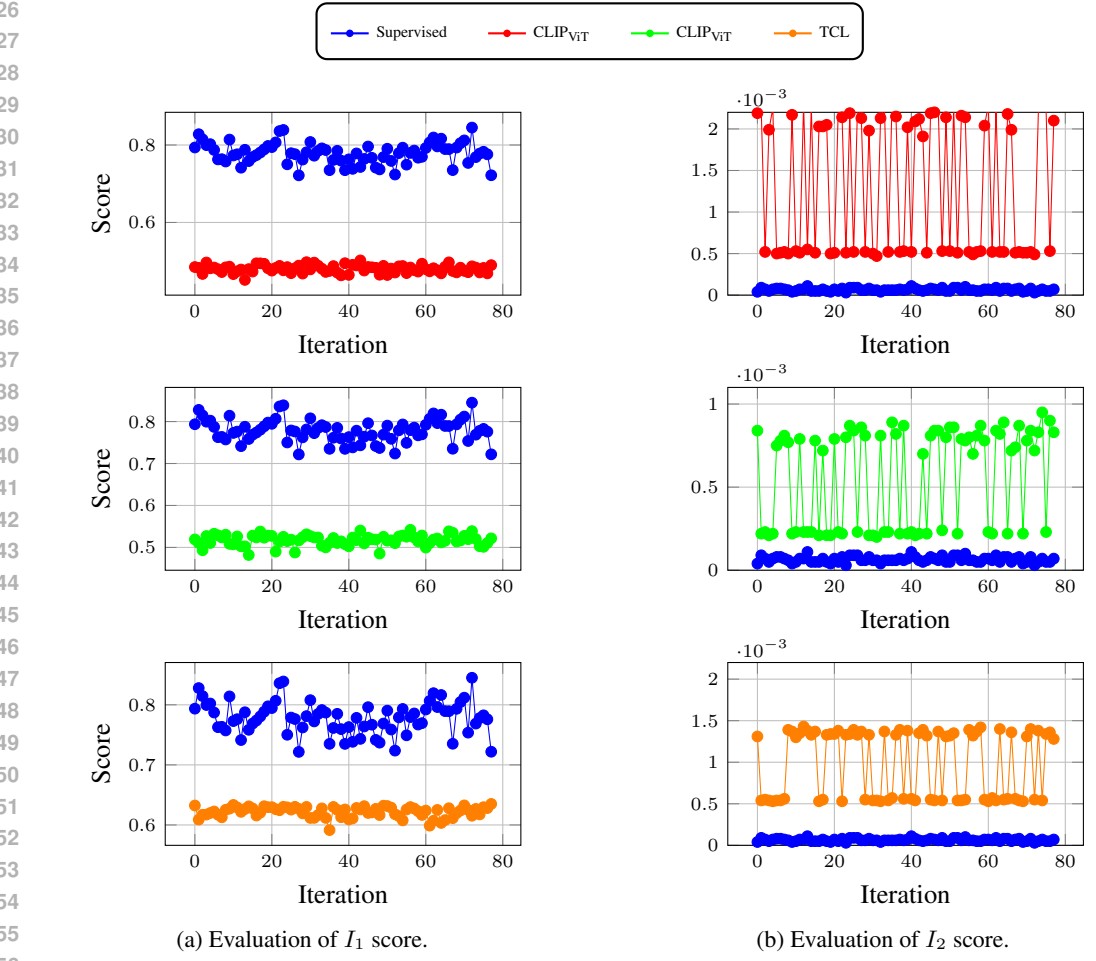

(a) Evaluation of $I_1$ score.

(b) Evaluation of $I_2$ score.

Figure 3: Comparison of isotropy metrics in VLPs vs. supervised model for ImageNet V2 data. Each row compares isotropy metrics for different vision-language pre-training methods (CLIP$_{\text{ViT}}$, ALBEF, TCL) alongside the supervised baseline.

Thus, the optimization problem can be formulated as:

$$\min_{z_i'} \frac{1}{2}\|z_i'\|_2^2, \quad \text{s.t.} \quad (z_i' - z_i)^\top \Sigma^{-1}(z_i' - z_i) \leq 1. \tag{17}$$

To solve this constrained optimization problem, we construct the Lagrangian function:

$$\mathcal{L}(z_i', \lambda) = z_i'^\top z_i' + \lambda\big((z_i' - z_i)^\top \Sigma^{-1}(z_i' - z_i) - 1\big), \quad \lambda \geq 0. \tag{18}$$

where $\lambda$ is the Lagrange multiplier. Differentiating the Lagrangian with respect to $z_i'$:

$$\frac{\partial \mathcal{L}}{\partial z_i'} = 2z_i' + \lambda \Sigma^{-1}(z_i' - z_i) = 0.$$

With simplifying, we have:

$$z_i'^* = (\Sigma + \lambda I)^{-1}\lambda z_i.$$

where $\lambda > 0$ is the regularization parameter. $\qquad \square$

### B.3 PROOF OF LEMMA 3.4

*Proof.* Let
$$\Sigma = U \operatorname{diag}(\sigma_1, \ldots, \sigma_D) U^\top$$
be the eigendecomposition of the covariance matrix, where eigenvectors $u_j$ with large $\sigma_j$ span the tangent space of $\mathcal{M}$ and those with small $\sigma_j$ span the normal space. By Lemma 3.3, the optimal adversarial embedding is
$$z_i'^* = (\Sigma + \lambda I)^{-1} \lambda z_i.$$
Substituting the spectral decomposition and projecting into the eigenbasis gives
$$U^\top z_i'^* = \operatorname{diag}\left(\frac{\lambda}{\sigma_j + \lambda}\right) U^\top z_i.$$
Define $\tilde{z}_i = U^\top z_i$ and $\tilde{z}_i' = U^\top z_i'^*$;

As $U \in \mathbb{R}^{D \times D}$ is an orthonormal matrix with $U^\top U = I$. Then for any point $z' \in \mathbb{R}^D$ and any set (manifold) $\mathcal{M} \subseteq \mathbb{R}^D$,
$$\inf_{m \in \mathcal{M}} \|z' - m\| = \inf_{m \in \mathcal{M}} \| U^\top z' - U^\top m\|. \tag{19}$$
Thus, projecting onto the eigenbasis via $U^\top$ exactly preserves the minimum distance from $z'$ to $\mathcal{M}$.

then
$$\tilde{z}_i' = \operatorname{diag}\left(\frac{\lambda}{\sigma_j + \lambda}\right) \tilde{z}_i.$$
By Assumption 3.1 (anisotropy), the spectrum satisfies $\sigma_j \gg \lambda$ for tangent directions and $\sigma_j \ll \lambda$ for normal directions. Hence
$$\frac{\lambda}{\sigma_j + \lambda} \begin{cases} \approx 0, & \text{if } \sigma_j \gg \lambda \quad \text{(tangent directions)}, \\ \approx 1, & \text{if } \sigma_j \ll \lambda \quad \text{(normal directions)}. \end{cases}$$
Clean embedding lies very close to the manifold, which means that almost all of its magnitude is in directions tangent to that manifold: the normal (off-manifold) component is so small that it can be ignored.

Conversely, the adversarial mapping derived in Lemma 3.3 effectively suppresses the tangent components (scaling them by a negligible factor) while preserving or even slightly amplifying the normal components. Consequently, the optimal adversarial point lies off the clean data manifold.

$\square$

### B.4 PROOF OF THEOREM 3.5

*Proof.* We begin by establishing a lower bound on the distance between a perturbed sample embedding $z_i'$ and a random clean sample embedding $z_u$. By the triangle inequality, the distance between $z_i'$ and a random $z_u$ satisfies:
$$\|z_i' - z_u\| \geq \|z_i' - \mathcal{M}\| - \|\mathcal{M} - z_u\|.$$

Using the Lemma 3.4, we have $\|z_i - \mathcal{M}\| \geq \gamma$, and by Assumption 3.2, we have $\|\mathcal{M} - z_u\| \leq \alpha$. Substituting these bounds into the inequality, we obtain:
$$\|z_i' - z_u\| \geq \gamma - \alpha.$$
Taking expectations over the distribution $z_u \sim p(z)$, the expected adversarial distance satisfies:
$$\mathbb{E}[Dist_a] = \mathbb{E}_{z_u \sim p}[\|z_i' - z_u\|] = \int \|z_i' - z_u\| p(z_u)\, du \geq \gamma - \alpha.$$

Now consider the clean embedding $z_i \sim p(\cdot)$, where both $z_i$ and $z_u$ lie within $\alpha$ from the manifold $\mathcal{M}$. Again, by the triangle inequality:
$$\|z_i - z_u\| \leq \|z_i - \mathcal{M}\| + \|\mathcal{M} - z_u\| \leq \alpha + \alpha = 2\alpha$$
Thus, we have: $\mathbb{E}[Dist_c] = \mathbb{E}_{z_u \sim p}[\|z_i - z_u\|] \leq 2\alpha.$

Finally, under the condition $\gamma > 3\alpha$, we obtain:
$$\mathbb{E}[Dist_a] \geq \gamma - \alpha > 3\alpha - \alpha = 2\alpha > \mathbb{E}[Dist_c]$$
$$\Rightarrow \mathbb{E}[Dist_a] > \mathbb{E}[Dist_c].$$
This completes the proof, demonstrating that the expected distance between a perturbed and clean sample, when $\gamma > 3\alpha$, exceeds that between two clean samples. $\square$

## B.5 IMPACT OF OFF-MANIFOLD SHIFT ON GEOMETRIC SCORES USING THEOREM 3.5

Using Lemma 3.4 and Theorem 3.5, we formalize how AEs cause geometric deviations. Let $z_u$ be a clean (on-manifold) satisfying $\|z_i - \mathcal{M}\| \le \alpha$, and let $z_i'$ be an adversarial (off-manifold) embedding such that satisfies $\|z_i' - \mathcal{M}\| \ge \gamma > 3\alpha$. From Theorem 3.5, we know:

$$\mathbb{E}_{z_u \sim p}\big[\|z_i' - z_u\|\big] \ > \ \mathbb{E}_{z_u \sim p}\big[\|z_i - z_u\|\big].$$

Using this difference, we show that standard distance- or density-based measures, including $k$, NN, KDE, Mahalanobis distance, and LID, will produce higher average scores (or lower, in the case KDE) for $z_i'$ than for $z_i$.

**$k$-NN Distance.** Let $k$-NN$(z_i) \subset \{z_1, \ldots, z_j\}$ be the set of $k$-nearest neighbors to $z_i$. Then we have

$$d_{\mathrm{kNN}}(z_i) \ = \ \frac{1}{k} \sum_{z_u \in \mathrm{kNN}(z_i)} \|z_i - z_u\|.$$

If $\|z_i' - z_u\|$ exceeds $\|z_i - z_u\|$ on average, then the set of candidate neighbors for $z_i'$ is at a larger radius. Formally, we have:

Let $r_k'$ be the distance from $z_i'$ to its $k$ nearest neighbors, and $r_k$ be the distance from $z_i$ to its $k$ nearest neighbors. By Theorem 3.5, we have $\mathbb{E}\big[\|z_i' - z_u\|\big] > \mathbb{E}\big[\|z_i - z_u\|\big]$ which implies (upon sorting distances from smallest to largest) that $\mathbb{E}[r_k'] > \mathbb{E}[r_k]$. Thus, we have:

$$\mathbb{E}\big[d_{\text{k-NN}}(z_i')\big] \ = \ \mathbb{E}\Big[\tfrac{1}{k} \sum_{z_u \in \text{k-NN}(z_i')} \|z_i' - z_u\|\Big] \ > \ \mathbb{E}\Big[\tfrac{1}{k} \sum_{z_u \in \text{k-NN}(z_i)} \|z_i - z_u\|\Big] \ = \ \mathbb{E}\big[d_{\text{k-NN}}(z_i)\big],$$

$$\Rightarrow \mathbb{E}[\, d_{\text{k-NN}}(z_i')\,] \ > \ \mathbb{E}[\, d_{\text{k-NN}}(z_i)\,].$$

**KDE.** Given i.i.d. clean samples $\{z_1, \ldots, z_n\}$, the kernel density estimate at $z_i$ is:

$$\hat{f}(z_i) \ = \ \frac{1}{n} \sum_{j=1}^{n} K\big(\|z_i - z_j\|\big).$$

where $K(\cdot)$ is a decreasing kernel function (e.g. Gaussian). From Theorem 3.5, for each $z_i$: $\|z_i' - z_j\| > \|z_i - z_k\|$, thus the term $K\big(\|z_i' - z_j\|\big)$ is *smaller* on average than $K(\|z_i - z_j\|)$ because $K$ is a monotonically decreasing function. Therefore,

$$\mathbb{E}\Big[K(\|z_i' - z_j\|)\Big] \ < \ \mathbb{E}\Big[K(\|z_i - z_j\|)\Big].$$

Summing over $j = 1, \ldots, n$ and dividing by $n$, we conclude

$$\mathbb{E}\big[\hat{f}(z_i')\big] \ < \ \mathbb{E}\big[\hat{f}(z_i)\big].$$

**Mahalanobis Distance.** Given a clean embedding distribution with mean $\mu \in \mathbb{R}^D$ and positive-definite covariance matrix $\Sigma \in \mathbb{R}^{D \times D}$, the Mahalanobis distance for a point $z_i \in \mathbb{R}^D$ is defined as:

$$d_{\mathrm{Mahal}}(z_i) \ = \ \sqrt{(z_i - \mu)^\top \Sigma^{-1} (z_i - \mu)}. \tag{20}$$

Suppose $z_i'$ is an off-manifold (adversarial) embedding as characterized by Lemma 3.4, and let $z_i$ be an on-manifold (clean) embedding, and then from Theorem 3.5, we know:

$$\mathbb{E}_{u \sim p}\big[\|z_i' - z_u\|\big] \ > \ \mathbb{E}_{u \sim p}\big[\|z_i - z_u\|\big].$$

Because $\Sigma \ne cI$, the Mahalanobis metric emphasizes directions of low variance (i.e., those with small eigenvalues of $\Sigma$). In particular, by Lemma 3.4, $z_i'$ lies in directions outside the principal

manifold, where $\Sigma$ typically has smaller eigenvalues (and hence larger eigenvalues of $\Sigma^{-1}$), leading to:

$$(z_i' - \mu)^\top \Sigma^{-1} (z_i' - \mu) \; > \; (z_i - \mu)^\top \Sigma^{-1} (z_i - \mu),$$

Taking expectation and then applying Jensen's inequality yields:

$$\mathbb{E}\Big[(z_i' - \mu)^\top \Sigma^{-1} (z_i' - \mu)\Big] \; > \; \mathbb{E}\Big[(z_i - \mu)^\top \Sigma^{-1} (z_i - \mu)\Big]$$

$$\Rightarrow \mathbb{E}\Big[\sqrt{(z_i' - \mu)^\top \Sigma^{-1}(z_i' - \mu)}\Big] \; > \; \mathbb{E}\Big[\sqrt{(z_i - \mu)^\top \Sigma^{-1}(z_i - \mu)}\Big].$$

reaching to:

$$\mathbb{E}\big[d_{\mathrm{Mahal}}(z_i')\big] \; > \; \mathbb{E}\big[d_{\mathrm{Mahal}}(z_i)\big].$$

**LID.** The maximum likelihood estimator (MLE) of LID for a point $z_i$ is given by:

$$\hat{\mathrm{LID}}(z_i) \; = \; \Big(-\frac{1}{k} \sum_{i=1}^{k} \log\Big[\frac{r_i(z_i)}{r_{\max}(z_i)}\Big]\Big)^{-1}.$$

where $r_j(z_j)$ is the distance to the $j$-th nearest neighbor and $r_{max}(z_i)$ is the distance to the farthest among the $k$ neighbors. By Lemma 3.4, adversarial embeddings tend to lie farther from the data manifold, resulting in increased distances:

$$r_i(z_i') = r_i(z_i) + \Delta r_i, \quad r_{\max}(z_i') = r_{\max}(z_i) + \Delta r_{\max},$$

where $\Delta r_i > 0$, $\Delta r_{\max} > 0$ are perturbation shifts.

Substitute these into the LID formula for $z_i'$:

$$\mathrm{LID}_{\mathrm{adv}}(z_i') = \left(-\frac{1}{k} \sum_{j=1}^{k} \log \frac{r_j(z_i) + \Delta r_j}{r_{\max}(z_i) + \Delta r_{\max}}\right)^{-1}.$$

We assume that adversarial perturbations enlarge the inner radius of the local neighbourhood more than the outer radii, as in highly anisotropic embedding spaces, variance concentrates along a few principal axes. Adversarial perturbations, causing data to be off-manifold, step into low-variance directions separate the perturbed point from its immediate neighbours, while barely affecting its distance to already distant points, reaching to:

$$\frac{\Delta r_j}{r_j(z_i)} \; > \; \frac{\Delta r_{\max}}{r_{\max}(z_i)} \quad \text{for all } j < k.$$

By simplifying the log difference and using the log difference property, we have:

$$\log\left(1 + \frac{\Delta r_j}{r_j(z_i)}\right) > \log\left(1 + \frac{\Delta r_{\max}}{r_{\max}(z_i)}\right) \Rightarrow \frac{1}{k} \sum_{j=1}^{k} \log \frac{r_j(z_i) + \Delta r_j}{r_{\max}(z_i) + \Delta r_{\max}} > \frac{1}{k} \sum_{i=1}^{k} \log \frac{r_j(z_i)}{r_{\max}(z_i)}.$$

$$(21)$$

$$\Rightarrow \mathrm{LID}_{adv}(z_i') > \mathrm{LID}(z_i). \tag{22}$$

Therefore, by taking the expectation:

$$\mathbb{E}[\mathrm{LID}_{adv}(z_i')] \; > \; \mathbb{E}[\mathrm{LID}(z_i)].$$

showing that the expectation of LID values of adversarial embeddings is higher than LID values of clean ones.

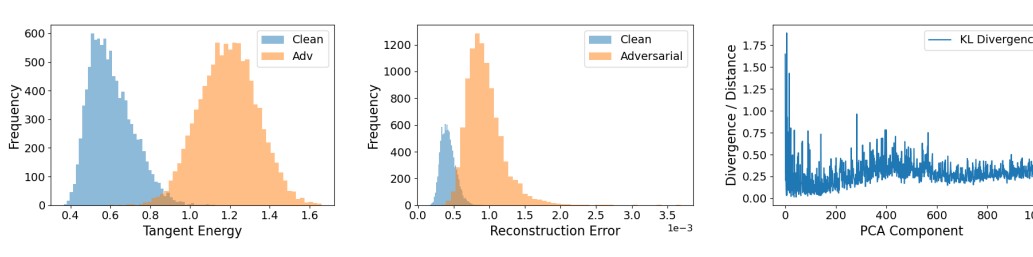

(a) Comparison of energy in normal direction.

(b) Comparison of reconstruction error.

(c) Divergence per PCA dimension (ordered by variance of clean data)

Figure 4: Verification of Lemma B.6: adversarial data in VLPs are off-manifold.

## B.6 OFF-MANIFOLD VERIFICATION (LEMMA 3.4)

To verify this phenomenon, we conducted four empirical validations to support this claim (using ImageNet-V2 and $CLIP_{CNN}$) based on different perspectives.

- **Energy in Normal Directions**: By decomposing the embedding space using SVD, in Figure 4a it is shown that adversarial samples exhibit significantly greater energy in the directions corresponding to the lowest 10 singular values, those least used by clean data, compared to clean samples. Since these directions span the normal space to the data manifold, this confirms that adversarial perturbations push samples off the manifold.

- **Reconstruction Error via Low-Rank PCA**: We projected both clean and adversarial embeddings onto the top K=10 principal components derived from clean data, and measured the reconstruction error (L2 residual). Figure 4b shows that adversarial samples have higher residuals, indicating they lie farther from the low-dimensional subspace characterizing clean data. This shows that adversarial examples activate uncommon directions and break the compact structure of the manifold.

- **KL Divergence in Principal Directions**: To assess the statistical shift, we computed KL divergence between clean and adversarial embeddings along individual PCA axes (up to 1000, ordered by variance on the clean data). Figure 4c shows that divergence is most pronounced in the top 1–20 components, suggesting that adversarial perturbations alter the most informative and discriminative dimensions of the representation space.

- **Instability of Principal Directions**: We further examined whether the top PCA directions remain stable under attack by computing Kendall's Tau (Kendall, 1938) and Spearman's Rho (Spearman, 1987) rank correlations between the top 10 clean and adversarial components. The correlations were consistently low, indicating that adversarial perturbations disrupt the structure of dominant semantic directions in the embedding space.

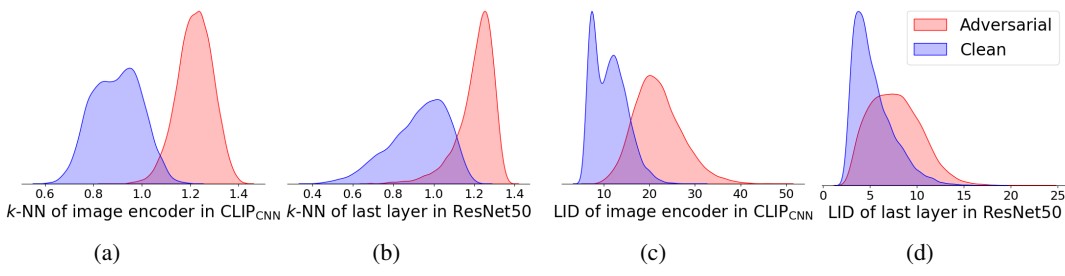

(a)        (b)        (c)        (d)

Figure 5: Comparison of the $k$-NN and LID distributions between the $CLIP_{CNN}$ image encoder and the ResNet-50 using ImageNet data.

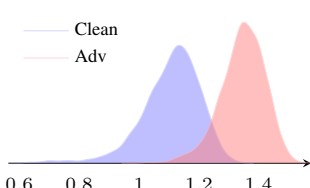

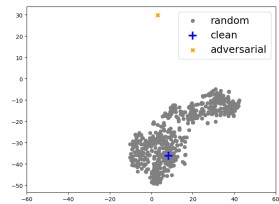

Figure 6: Evaluation of random sample distance distributions in $\text{CLIP}_{\text{CNN}}$ image encoder.

Figure 7: t-SNE visualization of $\text{CLIP}_{\text{CNN}}$ image features. Gray represents randomly selected CIFAR10 data, blue a clean sample, and orange its adversarial counterpart.

### B.7 EMPIRICAL VERIFICATION OF GEODETECT

In this section, we first present our motivation for using geometric metrics as signals to detect AEs in VLPs. We then empirically demonstrate that the geometric distance gap, as formalized in Theorem 3.5, serves as an effective and interpretable signal for adversarial detection.

We compare the distributions of $k$-NN distances and LID scores in $\text{CLIP}_{\text{CNN}}$ and a supervised ResNet-50 baseline, both sharing the same image encoder architecture. As shown in Figure 5, $\text{CLIP}_{\text{CNN}}$ exhibits a more pronounced separation between clean and adversarial samples in both metrics. This indicates that adversarial perturbations induce stronger geometric deviations in CLIP embeddings, making them easier to distinguish from clean samples. This observation aligns with our earlier analysis of anisotropy that the more directional, concentrated structure of VLPs embeddings leads to stronger off-manifold deviations under perturbation. These results provide empirical support for the theoretical foundation that anisotropy amplifies geometric separation, enhancing adversarial detectability in VLPs.

Figure 6 visualizes the distribution of Euclidean distances between a query embedding and a randomly sampled clean reference. Clean–clean pairs (blue) cluster at lower values, while clean–adversarial pairs (red) shift noticeably to the right, in agreement with Theorem 3.5 statement that $\mathbb{E}_{z_u \sim p(\cdot)}\big[\|z_i' - z_u\|\big] > \mathbb{E}_{z_u \sim p(\cdot)}\big[\|z_i - z_u\|\big]$.

Table 6 builds on Lemma 3.4, which shows that adversarial examples lie off the clean-data manifold, and on Theorem 3.5, which shows the resulting gap in expected distances. Motivated by this insight, and to verify that, we test a simple "random-$k$" detector, where each query embedding is evaluated based on its distance to a single, randomly chosen clean embedding. Despite its simplicity, this approach effectively exploits the Lemma 3.4's idea, providing strong empirical performance and confirming that even a single distance captures meaningful adversarial deviations.

Finally, Figure 7 shows the t-SNE projection of a random batch using $\text{CLIP}_{\text{CNN}}$. Gray points represent randomly sampled CIFAR-10 clean embeddings. The blue point corresponds to one of these clean samples, and the orange point is its adversarial counterpart. The adversarial embedding is visibly displaced from the dense cluster of clean data, further reinforcing the claim that adversarial perturbations cause off-manifold shifts that are detectable through distance-based metrics.

## C ABLATION STUDY: SENSITIVITY ANALYSIS

In this subsection, we evaluate the sensitivity of GeoDetect to the locality parameters, the effect of different backbone attacks, and excluding multimodal embedding.

### C.1 SENSITIVITY TO LOCALITY

Adversarial detection methods based on local analysis, such as $k$-NN and LID, are sensitive to the locality hyperparameter $k$, which defines their local scope. To assess the impact of this parameter, we conducted experiments varying $k$ over $10, 20, 30, 40, 50$ using the $\text{Sep}_{\text{uni}}$ attack on $\text{CLIP}_{\text{ViT}}$. Figure 8 illustrates that $k$-NN exhibits more stable detection performance across different choices of

Table 6: Discrimination power (AUC and FPR95) of the random-$k$ distance for CLIP$_{\text{CNN}}$ and ALBEF across different datasets and attack types.

| Model | Attack | CIFAR10 | | CIFAR100 | | ImageNet1k | | STL10 | | Food101 | |
|---|---|---|---|---|---|---|---|---|---|---|---|
| | | AUC | FPR95 | AUC | FPR95 | AUC | FPR95 | AUC | FPR95 | AUC | FPR95 |
| CLIP$_{\text{CNN}}$ | Sep$_{\text{uni}}$ | 100 | 0.0 | 100 | 0.0 | 99.83 | 0.44 | 99.99 | 0.0 | 100 | 0.0 |
| | Co-Attack | 100 | 0.0 | 100 | 0.0 | 99.93 | 0.20 | 99.99 | 0.0 | 99.99 | 0.0 |
| CLIP$_{\text{ViT}}$ | Sep$_{\text{uni}}$ | 99.99 | 0.0 | 100 | 0.0 | 99.97 | 0.05 | 100 | 0.0 | 99.99 | 0.0 |
| | Co-Attack | 100 | 0.0 | 100 | 0.0 | 98.50 | 76.24 | 99.99 | 0.0 | 99.99 | 0.0 |
| ALBEF | Sep$_{\text{uni}}$ | 100 | 0.0 | 99.99 | 0.0 | 99.32 | 3.34 | 99.97 | 0.06 | 99.96 | 0.12 |
| | Sep$_{\text{multi}}$ | 99.88 | 0.39 | 99.43 | 2.90 | 65.65 | 88.93 | 76.99 | 76.25 | 88.34 | 42.66 |
| | Co-Attack | 100 | 0.0 | 99.99 | 0.0 | 99.29 | 3.74 | 99.95 | 0.06 | 99.95 | 0.10 |
| TCL | Sep$_{\text{uni}}$ | 100 | 0.0 | 100 | 0.0 | 94.80 | 23.27 | 99.99 | 0.0 | 99.98 | 0.0 |
| | Sep$_{\text{multi}}$ | 99.95 | 0.20 | 99.94 | 0.20 | 47.16 | 97.15 | 86.58 | 57.36 | 94.11 | 23.74 |
| | Co-Attack | 100 | 0.0 | 100 | 0.0 | 94.69 | 24.59 | 99.99 | 0.0 | 99.98 | 0.02 |

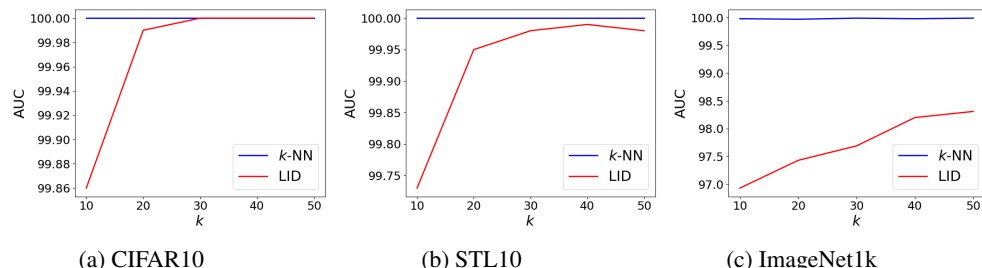

(a) CIFAR10         (b) STL10         (c) ImageNet1k

Figure 8: The detection AUC rates of local geometric approaches under varying locality k.

$k$, highlighting its robustness. In contrast, LID performance fluctuates more substantially, indicating greater sensitivity to neighborhood size selection.

## C.2 SENSITIVITY TO BATCH SIZE

Also, Table 7 illustrates the effect of different batch sizes on detection performance (AUC) for CLIP$_{\text{CNN}}$ using the ImageNet dataset, under the Sep-attack with the number of nearest neighbors ($k$) set to 20. As shown, the results remain relatively stable across batch sizes, indicating that detection performance is not significantly impacted. This suggests that even with a limited batch size of 64 and 20 nearest neighbors, GeoDetect remains highly effective at identifying adversarial samples.

Table 7: Effect of batch size on detection performance (AUC) with $k = 20$ nearest neighbors on ImageNet under Sep-attack.

| Batch Size | GeoDetect-$k$NN (%) | GeoDetect-LID (%) |
|---|---|---|
| 64 | 99.62 | 97.09 |
| 128 | 99.66 | 97.73 |
| 256 | 99.61 | 97.11 |
| 512 | 99.63 | 97.84 |

## C.3 SENSITIVITY TO LAYER SELECTION IN LID-BASED DETECTION

Table 8 reports the sensitivity of GeoDetect based on LID to detect Sep-Attack using different ImageNet layers. The results justify our choice to combine early, mid, and late blocks (Appendix A.3), as this captures complementary semantic depths that are necessary for effective detection.

Table 8: GeoDetect-LID performance (AUC %) using features from different layers/blocks in $\text{CLIP}_{\text{ViT}}$ using ImageNet.

| Layer / Block | GeoDetect-LID AUC (%) |
|---|---|
| Only Early | 64.64 |
| Only Mid | 62.92 |
| Only Late | 98.93 |
| Early + Mid | 65.67 |
| Mid + Late | 98.99 |
| Early + Mid + Late | 99.22 |

### C.4 SENSITIVITY TO NUMBER OF AVAILABLE SAMPLES

Table 9 reports the impact of the number of available samples $N$ on training the GeoDetect-LID detector for ImageNet with $\text{CLIP}_{\text{ViT}}$. The results show that performance remains stable across different sample sizes. For $k$-NN–based detection, both AUC and FPR95 are unaffected by $N$, since the GeoDetect $k$-NN is threshold-based.

Table 9: Impact of $N$ on GeoDetect-LID performance (AUC) for ImageNet using $\text{CLIP}_{\text{ViT}}$.

| $N/2$ | GeoDetect-LID (%) |
|---|---|
| 8000 | 99.22 |
| 7000 | 99.20 |
| 6000 | 99.16 |
| 5000 | 99.15 |

### C.5 THE EFFECT OF MULTIMODAL EMBEDDINGS ON DETECTION.

We evaluate the impact of including a multimodal layer on LID detection. As shown in Figure 9, the exclusion of this layer led to decreased AUC scores across most evaluated datasets. This highlights the critical role of multimodal embeddings, particularly when adversarial perturbations directly target the multimodal encoder.

## D ABLATION STUDY: GENERALIZATION ANALYSIS

### D.1 GENERALIZATION TO DIFFERENT BACKBONE ATTACKS.

We conducted an evaluation to assess GeoDetect's ability to generalize to new attacks. beyond PGD-based attacks. Specifically, for LID-based detection, we trained the detector using PGD-based attacks and then evaluated its performance against samples generated from other attack strategies, including FGSM (Goodfellow et al., 2014), R-FGSM (Tramèr et al., 2018), I-FGSM (Kurakin et al., 2018), and MI-FGSM (Dong et al., 2018). The calibration and test sets were prepared consistently with previous experiments, with calibration data exclusively using PGD-generated examples, while alternative attacks were reserved for the test set. For threshold-based methods ($k$-NN, Mahalanobis, and KDE), we directly evaluated their performance against these new attack strategies without additional training. Results presented in Table 10 demonstrate substantial generalizability and robustness across diverse gradient-based adversarial attacks.

### D.2 EVALUATION OF ROBUSTNESS AGAINST SET-LEVEL GUIDANCE (SGA) ATTACK

In this subsection, we evaluate the robustness of GeoDetect in detecting SGA attack. SGA (Lu et al., 2023) is a strong VLP-specific adversarial attack that perturbs inputs by optimizing against a set of semantically irrelevant text prompts. This strategy effectively disrupts image-text alignment across multiple candidate prompts, making it harder to detect using conventional pairwise similarity. In

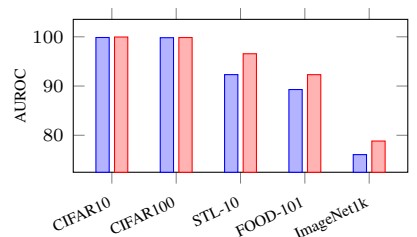

Figure 9: Effect of multimodal embedding on LID detection under fused VLP attacks.

Table 10: Generalization of GeoDetect to $Sep_{uni}$ with different baseline attacks in $CLIP_{ViT}$ for STL10. AUC is reported.

| Attack | Method | | | |
|--------|--------|-------|--------|-------|
| | LID | $k$-NN | Mahal. | KDE |
| PGD | 99.99 | 100 | 99.99 | 99.97 |
| FGSM | 91.93 | 98.99 | 99.59 | 99.12 |
| R-FGSM | 75.39 | 74.12 | 94.11 | 83.91 |
| I-FGSM | 99.99 | 100 | 99.99 | 99.96 |
| MI-FGSM | 99.97 | 100 | 99.99 | 99.96 |

this attack, we use a batch size of 64 for $CLIP_{CNN}$ and $CLIP_{ViT}$, with $k = 40$ for LID and $k = 5$ for $k$-NN. For ALBEF and TCL, the batch size is set to 16, with $k = 10$ for both LID and $k$-NN.

Table 11: GeoDetect discrimination power (AUC score) comparison of SGA attack with Co-Attack in Image-Retrieval Task with Flickr30k and COCO dataset in aligned VLPs ($CLIP_{CNN}$ and $CLIP_{ViT}$), and fused VLPs (ALBEF and TCL)

(a) $k$-NN Detection

| Model | Flickr30k | | COCO | |
|-------|-----------|-----|------|-----|
| | Co-Attack | SGA | Co-Attack | SGA |
| $CLIP_{CNN}$ | 99.97 | 93.89 | 99.95 | 90.78 |
| $CLIP_{ViT}$ | 99.59 | 94.41 | 99.51 | 91.75 |
| ALBEF | 99.88 | 82.71 | 98.73 | 84.15 |
| TCL | 96.59 | 67.32 | 98.05 | 91.11 |

(b) LID Detection

| Model | Flickr30k | | COCO | |
|-------|-----------|-----|------|-----|
| | Co-Attack | SGA | Co-Attack | SGA |
| $CLIP_{CNN}$ | 98.90 | 88.18 | 99.50 | 91.97 |
| $CLIP_{ViT}$ | 96.55 | 85.47 | 99.06 | 86.00 |
| ALBEF | 93.80 | 73.75 | 91.49 | 75.12 |
| TCL | 90.76 | 70.12 | 88.25 | 78.58 |

The GeoDetect's discrimination performance against the SGA attack, as presented in Table 11, demonstrates a discrimination power comparable to that of Co-Attack. Moreover, even under stronger attack scenarios, GeoDetect achieves a robust detection rate.

### D.2.1 SGA IN BLACK-BOX SETTINGS

We evaluated the SGA attack on the Flickr30k dataset to assess GeoDetect's robustness under black-box conditions. As reported in Table 12, GeoDetect-LID is able to detect transfer attacks.

Table 12: GeoDetect-LID AUC (%) under black-box SGA transfer attacks on Flickr30k.

| Source $\rightarrow$ Target | GeoDetect-LID AUC (%) |
|-----------------------------|------------------------|
| CLIP(CNN) $\rightarrow$ CLIP(ViT) | 88.08 |
| CLIP(ViT) $\rightarrow$ CLIP(CNN) | 82.32 |

### D.3 EVALUATION OF ADAPTIVE ATTACKS

In this section, we evaluate the impact of adaptive attacks at test time on GeoDetect. Adaptive attacks specifically target the detection mechanism by incorporating it into the optimization process for perturbation generation. Assessing their effectiveness is critical, particularly in white-box settings, where the attacker has full access to the model and can modify the optimization function to craft perturbations that directly target the detection method. Additionally, we categorize the adaptive attacks into two groups based on the method of perturbation generation and evaluate the performance of GeoDetect against each type of adaptive attack. In the following attack setting, the batch size for $CLIP_{CNN}$ and $CLIP_{ViT}$ is set to 128, with $k = 100$ for LID and $k = 10$ for $k$-NN. For ALBEF and TCL, the batch size is set to 32, with $k = 20$ for LID and $k = 10$ for $k$-NN.

Table 13: GeoDetect discrimination power (AUC score) comparison between different distribution adaptive and non-adaptive attacks for Image-Retrieval Task with Flickr30k and COCO dataset in aligned VLPs ($CLIP_{CNN}$ and $CLIP_{ViT}$), and fused VLPs (ALBEF and TCL) (Note: 'N-adaptive' refers to the Non-adaptive method.)

(a) Effect of $k$-NN adaptive Attacks

| Model | Attack | Dataset | | | |
|---|---|---|---|---|---|
| | | Flickr30k | | COCO | |
| | | N-adaptive | Adaptive | N-adaptive | Adaptive |
| $CLIP_{CNN}$ | $Sep_{uni}$ | 99.99 | 67.82 | 99.97 | 74.47 |
| | Co-Attack | 99.97 | 67.32 | 99.95 | 74.28 |
| $CLIP_{ViT}$ | $Sep_{uni}$ | 100 | 61.16 | 100 | 68.44 |
| | Co-Attack | 99.59 | 59.68 | 99.51 | 67.68 |
| ALBEF | $Sep_{uni}$ | 99.75 | 51.38 | 98.54 | 71.53 |
| | $Sep_{multi}$ | 54.84 | 49.82 | 57.02 | 69.88 |
| | Co-Attack | 99.88 | 51.02 | 98.73 | 71.53 |
| TCL | $Sep_{uni}$ | 96.10 | 51.27 | 98.01 | 74.19 |
| | $Sep_{multi}$ | 32.89 | 50.57 | 33.89 | 74.04 |
| | Co-Attack | 96.59 | 51.18 | 98.05 | 74.00 |

(b) Effect of LID adaptive Attacks

| Model | Attack | Dataset | | | |
|---|---|---|---|---|---|
| | | Flickr30k | | COCO | |
| | | N-adaptive | Adaptive | N-adaptive | Adaptive |
| $CLIP_{CNN}$ | $Sep_{uni}$ | 98.45 | 83.81 | 99.54 | 93.14 |
| | Co-Attack | 98.90 | 82.81 | 99.50 | 94.67 |
| $CLIP_{ViT}$ | $Sep_{uni}$ | 99.37 | 31.56 | 99.98 | 83.82 |
| | Co-Attack | 96.55 | 52.24 | 99.06 | 86.96 |
| ALBEF | $Sep_{uni}$ | 94.99 | 71.78 | 91.80 | 89.86 |
| | $Sep_{multi}$ | 74.26 | 89.83 | 79.85 | 92.31 |
| | Co-Attack | 93.80 | 70.12 | 91.49 | 90.11 |
| TCL | $Sep_{uni}$ | 90.88 | 77.15 | 89.32 | 88.70 |
| | $Sep_{multi}$ | 84.72 | 91.83 | 83.95 | 94.64 |
| | Co-Attack | 90.76 | 78.10 | 88.25 | 90.02 |

**Different Distribution Adaptive Attacks**  In this subsection, we assess the effect of adaptive attacks, where the batch used to generate the attack differs from the batch used for detection. This ensures that the attack is strong enough to evade the specific batch distribution. This approach challenges detection methods to generalize across attacks crafted from different data distributions, thereby enhancing the credibility of the results and demonstrating the practical resilience of the detection framework. One of the challenges in generating AEs is that minimizing the gradient of the distance to the current k-nearest neighbors is not always representative of the true direction for optimizing the set of k-nearest neighbors (Athalye et al., 2018). Our approach of attacking mitigates this issue by selecting different batches for attack generation and detection. By doing so, we avoid optimizing based on the current nearest neighbors in the same batch, which could mislead the attack's effectiveness. This method ensures that the attack is strong enough to bypass detection while maintaining the true nature of adversarial perturbations.

Moreover, GeoDetect based on $k$-NN is inherently robust due to its threshold-based detection, and the adversary has no access to the detector model to evade it.

$$\mathcal{L}_{adaptive}(z_i, z_i') = \mathcal{L}_{main}(z_i, z_i') - \zeta \cdot \text{Metric}(z_i', \{z_j\}_{j=1}^n), z_i \in B_g \neq B_d. \tag{23}$$

Here, $\text{Metric}(z_i', \{z_j\}_{j=1}^n)$ represents the LID or $k$-NN function that computes the score for AE embeddings $z_i'$ relative to the clean sample embeddings $z_i$. The batch used for attack generation, $B_g$, is different from the batch used for detection, $B_d$. We set $\zeta = 0.1$, $k = 20$ for LID, and $k = 10$ for $k$-NN in optimization of attacks. The results presented in Table 13 offer insights into the resilience of the GeoDetect framework, especially GeoDetect-LID framework, under adaptive attacks. $k$-NN and LID are robust when detecting adaptive attacks for CLIP and reasonably effective for ALBEF and TCL. We also observe that LID detection is more robust to adaptive attacks, highlighting its effectiveness and capability in more complex types of attack. The improved detection rates, especially against $Sep_{multi}$ attacks, can be attributed to the dynamics of attack generation. Specifically, the use of a multimodal encoder during attack generation alters the data distribution, enhancing the distinguishability of perturbed samples. This process causes the perturbed samples to shift more significantly within the feature space, making them easier to detect.

**Selective Gradient Descent Adaptive Attack**  In this section, we evaluate the selected adaptive attack introduced by Bryniarski et al. (2021), which addresses the issue of over-optimization when one objective is prioritized at the expense of another. This attack method ensures a more balanced optimization process, avoiding the trade-off that could compromise the effectiveness of adversarial perturbations in evading detection while still achieving misclassification. The paper argues that summing the misclassification loss $\mathcal{L}_{adaptive}(z_i, z_i')$ and detection loss $\text{Metric}(z_i', \{z_j\}_{j=1}^n)$ is problematic because the two terms represent conflicting objectives. The misclassification term aims

Table 14: GeoDetect discrimination power (AUC score) comparison between selective gradient descent adaptive and non-adaptive attacks for Image-Retrieval Task with Flickr30k and COCO dataset in aligned VLPs (CLIP$_{\text{CNN}}$ and CLIP$_{\text{ViT}}$), and fused VLPs (ALBEF and TCL) (Note: 'N-adaptive' refers to the Non-adaptive method.)

(a) Effect of $k$-NN adaptive Attacks

| Model | Attack | Dataset | | | |
|---|---|---|---|---|---|
| | | Flickr30k | | COCO | |
| | | N-adaptive | Adaptive | N-adaptive | Adaptive |
| CLIP$_{\text{CNN}}$ | Sep$_{\text{uni}}$ | 99.99 | 67.97 | 99.97 | 74.17 |
| | Co-Attack | 99.97 | 67.75 | 99.95 | 74.93 |
| CLIP$_{\text{ViT}}$ | Sep$_{\text{uni}}$ | 100 | 60.97 | 100 | 68.73 |
| | Co-Attack | 99.59 | 61.38 | 99.51 | 68.23 |
| ALBEF | Sep$_{\text{uni}}$ | 99.75 | 52.01 | 98.54 | 71.28 |
| | Sep$_{\text{multi}}$ | 54.84 | 49.93 | 57.02 | 69.80 |
| | Co-Attack | 99.88 | 51.40 | 98.73 | 70.94 |
| TCL | Sep$_{\text{uni}}$ | 96.10 | 51.26 | 98.01 | 74.08 |
| | Sep$_{\text{multi}}$ | 32.89 | 50.77 | 33.89 | 74.23 |
| | Co-Attack | 96.59 | 50.91 | 98.05 | 73.85 |

(b) Effect of LID adaptive Attacks

| Model | Attack | Dataset | | | |
|---|---|---|---|---|---|
| | | Flickr30k | | COCO | |
| | | N-adaptive | Adaptive | N-adaptive | Adaptive |
| CLIP$_{\text{CNN}}$ | Sep$_{\text{uni}}$ | 98.45 | 86.10 | 99.54 | 93.50 |
| | Co-Attack | 98.90 | 82.72 | 99.50 | 94.75 |
| CLIP$_{\text{ViT}}$ | Sep$_{\text{uni}}$ | 99.37 | 35.83 | 99.98 | 86.24 |
| | Co-Attack | 96.55 | 62.98 | 99.06 | 88.88 |
| ALBEF | Sep$_{\text{uni}}$ | 94.99 | 76.34 | 91.80 | 90.86 |
| | Sep$_{\text{multi}}$ | 74.26 | 88.68 | 79.85 | 92.78 |
| | Co-Attack | 93.80 | 74.92 | 91.49 | 90.92 |
| TCL | Sep$_{\text{uni}}$ | 90.88 | 78.90 | 89.32 | 89.38 |
| | Sep$_{\text{multi}}$ | 84.72 | 91.93 | 83.95 | 94.76 |
| | Co-Attack | 90.76 | 80.65 | 88.25 | 89.78 |

to fool the model, while the detection loss works to avoid detection. Optimizing both objectives simultaneously in a single summed loss function leads to difficulties in finding a global minimum, as the problem is non-convex. This increases the likelihood of falling into local minima that focus too much on one objective and neglect the other, resulting in attacks that either fail to mislead the model or are easily detected. The formula for this adaptive attack is presented as follows:

$$\mathcal{L}_{\text{adaptive}}(z_i, z_i') = \mathcal{L}(z_i, z_i') \cdot \mathbb{1}[\text{Sim}(z_I, z_T) = t_{id}] - \rho \cdot \text{Metric}(z_i', \{z_j\}_{j=1}^n) \cdot \mathbb{1}[\text{Sim}(z_I, z_T) \neq t_{id}] \tag{24}$$

where similarity matrix defined in Equation 25, and $t_{id}$ represents the text IDs associated with the images. Also, we put $\rho = 0.1$ in our experiments, The core idea is that, instead of minimizing a convex combination of the two loss functions, the approach selectively optimizes either $\mathcal{L}_{\text{main}}$ or $\text{Metric}(z_i', \{z_j\}_{j=1}^n)$ based on whether the maximum similarity score corresponds to the text ID.

$$\text{Sim}(z_I, z_t) = \sum_{k=1}^n z_I^k \cdot z_T^k \tag{25}$$

The results presented in Table 14 offer insights into the resilience of the GeoDetect, especially GeoDetect-LID framework, under this type of adaptive attack.

# E  EXTENDED EVALUATION

## E.1  EVALUATION ON DIFFERENT MODELS

Table 15 presents the results of adversarial detection for zero-shot classification in the CLIP$_{\text{ViT}}$ and TCL models. The findings are consistent with those shown in Table 1 in the main text.

## E.2  COMPARISON WITH PROMPT-BASED IRRELEVANT PROBING

PIP (Prompt-based Irrelevant Probing) (Zhang et al., 2024c) is a task-specific detection method designed for VQA. It relies on analyzing the attention patterns in response to irrelevant probe questions to identify AEs. However, its applicability is limited to multimodal models that expose cross-attention layers between vision and language components. Specifically, PIP is not compatible with dual encoder architectures that lack explicit cross-attention mechanisms. Furthermore, since PIP operates on question-conditioned attention maps, it is inherently constrained to VQA and cannot be easily extended to other vision-language tasks or modalities. Its detection strategy is also not

Table 15: A comparison of the discrimination power (AUC score) among MCM and GeoDetect framework using LID, $k$-NN, Mahalanobis (denoted as Mah.) and KDE in an aligned VLP, CLIP$_\text{ViT}$, and a fused VLPs, TCL.

| Model | Method | Attack | CIFAR10 | | CIFAR100 | | ImageNet1k | | STL10 | | Food101 | |
|---|---|---|---|---|---|---|---|---|---|---|---|---|
| | | | AUC | FPR95 | AUC | FPR95 | AUC | FPR95 | AUC | FPR95 | AUC | FPR95 |
| CLIP$_\text{ViT}$ | MCM | Sep$_\text{uni}$ | 76.47 | 88.24 | 72.09 | 67.83 | 86.06 | 54.55 | 94.84 | 26.79 | 94.51 | 25.32 |
| | | Co-Attack | 80.21 | 73.54 | 68.37 | 76.24 | 84.14 | 58.93 | 95.51 | 20.73 | 89.78 | 40.95 |
| | LID | Sep$_\text{uni}$ | 100 | 0.00 | 100 | 0.00 | 99.23 | 4.57 | 99.99 | 0.00 | 99.98 | 0.02 |
| | | Co-Attack | 100 | 0.00 | 100 | 0.00 | 97.09 | 15.30 | 99.74 | 0.64 | 99.54 | 1.49 |
| | $k$-NN | Sep$_\text{uni}$ | 100 | 0.00 | 100 | 0.00 | 99.98 | 0.00 | 100 | 0.00 | 100 | 0.00 |
| | | Co-Attack | 100 | 0.00 | 100 | 0.00 | 98.67 | 6.64 | 99.99 | 0.00 | 100 | 0.00 |
| | Mah. | Sep$_\text{uni}$ | 100 | 0.00 | 100 | 0.00 | 99.85 | 0.94 | 99.99 | 0.06 | 99.98 | 0.14 |
| | | Co-Attack | 100 | 0.00 | 100 | 0.00 | 99.18 | 3.07 | 99.97 | 0.06 | 99.82 | 0.82 |
| | KDE | Sep$_\text{uni}$ | 100 | 0.0 | 100 | 0.00 | 99.95 | 0.10 | 99.97 | 0.19 | 100 | 0.00 |
| | | Co-Attack | 100 | 0.00 | 100 | 0.00 | 98.79 | 6.35 | 99.82 | 0.39 | 100 | 0.0 |
| TCL | MCM | Sep$_\text{uni}$ | 76.91 | 55.63 | 62.15 | 73.78 | 90.49 | 32.02 | 94.82 | 18.45 | 76.76 | 71.88 |
| | | Sep$_\text{multi}$ | 46.63 | 99.06 | 37.01 | 97.24 | 64.85 | 87.65 | 73.34 | 77.49 | 46.94 | 95.84 |
| | | Co-Attack | 80.82 | 45.65 | 69.05 | 68.32 | 92.74 | 26.71 | 97.13 | 13.65 | 79.07 | 64.75 |
| | LID | Sep$_\text{uni}$ | 100 | 0.00 | 100 | 0.00 | 91.92 | 28.28 | 99.62 | 1.81 | 99.77 | 1.01 |
| | | Sep$_\text{multi}$ | 99.91 | 0.25 | 99.88 | 0.54 | 85.64 | 53.66 | 97.71 | 12.97 | 93.15 | 30.67 |
| | | Co-Attack | 100 | 0.00 | 100 | 0.00 | 91.02 | 30.64 | 99.89 | 0.69 | 99.79 | 0.79 |
| | $k$-NN | Sep$_\text{uni}$ | 100 | 0.00 | 100 | 0.00 | 93.99 | 25.77 | 99.97 | 00.06 | 99.98 | 00.06 |
| | | Sep$_\text{multi}$ | 99.78 | 1.28 | 99.87 | 0.59 | 32.54 | 97.98 | 85.39 | 56.04 | 92.08 | 32.53 |
| | | Co-Attack | 100 | 0.00 | 100 | 0.00 | 93.83 | 26.61 | 99.98 | 00.06 | 99.98 | 0.06 |
| | Mah. | Sep$_\text{uni}$ | 100 | 0.00 | 100 | 0.00 | 99.65 | 1.56 | 99.99 | 0.06 | 100 | 0.00 |
| | | Sep$_\text{multi}$ | 100 | 0.00 | 99.99 | 0.05 | 59.92 | 89.52 | 98.28 | 8.31 | 99.11 | 3.41 |
| | | Co-Attack | 100 | 0.00 | 100 | 0.00 | 99.64 | 1.56 | 99.99 | 0.06 | 100 | 0.00 |
| | KDE | Sep$_\text{uni}$ | 99.16 | 0.86 | 100 | 0.00 | 90.89 | 35.33 | 98.63 | 4.56 | 99.95 | 0.24 |
| | | Sep$_\text{multi}$ | 98.97 | 1.46 | 99.80 | 1.11 | 59.70 | 85.08 | 80.72 | 60.62 | 92.30 | 33.70 |
| | | Co-Attack | 99.15 | 0.86 | 100 | 0.00 | 90.98 | 36.54 | 98.62 | 4.56 | 99.96 | 0.20 |

Table 16: Performance comparison of PIP and GeoDetect under different clean/adversarial sample ratios.

| Method | $\mathcal{M}_\text{clean}/\mathcal{M}_\text{adv}$ | Precision | Accuracy | F1-score |
|---|---|---|---|---|
| PIP | 1000 / 1000 | 90.91 | 95.00 | 95.24 |
| | 1000 / 100 | 50.00 | 90.91 | 66.67 |
| GeoDetect ($k$-NN) | 1000 / 1000 | 97.99 | 97.65 | 97.64 |
| | 1000 / 100 | 83.19 | 99.09 | 90.41 |
| GeoDetect (LID) | 1000 / 1000 | 94.66 | 90.95 | 90.56 |
| | 1000 / 100 | 65.73 | 95.00 | 77.32 |

theoretically grounded, and its performance depends heavily on the nature of the probe questions and attack specificity to VQA outputs.

In contrast, GeoDetect is a general-purpose adversarial detection framework that does not rely on attention mechanisms or task-specific structures. It utilizes model representations and incorporates geometric reasoning to detect off-manifold behavior in AEs. GeoDetect is task-agnostic, theoretically justified, and shown to generalize across various settings and models.

We compare GeoDetect with PIP in Table 16, using the COCO dataset and AEs targeting CLIP$_\text{ViT}$, followed by the pipeline in (Zhang et al., 2024c). For GeoDetect, $k$-NN is set $k$ to 5, and for LID is to 20, both with a batch size of 128. To have a fair comparison, for both $k$-NN and LID, we used the mean-pooled final-layer embeddings to compute metrics. Also, to compute the precision, accuracy,

and F1-score we set the threshold for detection as 0.65 for $k$-NN-based, and 0.32 for LID-based. Following (Zhang et al., 2024c), the PGD attack encompasses a 20-step iteration, with a step learning rate of 2/255 and an overall perturbation limit of $\epsilon_\infty = 8/255$. GeoDetect significantly outperforms PIP across all evaluation metrics. In particular, GeoDetect-$k$NN achieves the best overall performance, with high precision, accuracy, and F1-score even under imbalanced settings. GeoDetect-LID also performs competitively in balanced scenarios, highlighting the method's robustness and adaptability across detection strategies.

# F    USAGE OF LARGE LANGUAGE MODELS.

We used OpenAI's GPT-4 and GPT-5 models, with a limited capacity, for language editing and polishing of the manuscript text. The models were not involved in developing ideas, designing methods, and analyzing results.

