# OpenReview forum: "GeoDetect: Geometric Adversarial Detection for VLPs"
_ICLR.cc/2026/Conference — Submitted to ICLR 2026_

### Official Review · Reviewer_zP31 · 2025-10-15

**Soundness:** 2
**Presentation:** 3
**Contribution:** 2
**Rating:** 4
**Confidence:** 4

**Summary:**

This paper presents GeoDetect, a theoretically grounded and effective framework for detecting adversarial examples in vision-language pre-trained models (VLPs). The authors uncover geometric anisotropy in VLP embeddings and leverage it for robust detection. Experiments across multiple datasets and models show excellent results (AUC≈1.0). The paper is well-motivated and clearly written, though more analysis on computational cost and real-world scalability would strengthen it.

**Strengths:**

1.	GeoDetect is grounded in a strong theoretical analysis of the geometric structure of vision-language model embeddings, demonstrating that adversarial examples tend to deviate from the data manifold. This provides clear interpretability and a robust theoretical basis for the proposed detection approach.
2.	The method is task-agnostic and does not rely on specific network architectures or attention mechanisms, making it applicable to a wide range of vision-language models and downstream tasks with strong generalization ability.
3.	GeoDetect operates directly on existing model embeddings without additional training or parameter tuning, offering a lightweight and easily deployable solution with low computational overhead.

**Weaknesses:**

1.	Although presented as efficient, the computation of geometric measures such as k-NN or KDE in high-dimensional embedding spaces can be resource-intensive, limiting scalability for large-scale or real-time applications.
2.	While GeoDetect supports multimodal data in theory, the experiments mainly focus on image perturbations, with limited evaluation on joint image–text adversarial attacks, leaving its robustness against complex cross-modal attacks less explored.
3.	Although the analysis of embedding-space anisotropy is carefully conducted, the findings are somewhat expected — prior work has already hinted that adversarial examples tend to move off the data manifold. Thus, while the theoretical framing is sound, the insight is not particularly surprising.

**Questions:**

Please see weakness

---

> ### Author Response · Authors · 2025-11-21
> **Response to Reviewer zP31**
>
> Thank you for taking the time to review our paper and providing valuable feedback. Below, you will find our responses to your questions:
>
> > **W1.** Computational efficiency and scalability
>
> Empirically, Appendix A.4 and Table 3 show that all metrics except LID require only 30–100 seconds, and LID takes ~9 minutes for a full CIFAR-10 run on a single H100 GPU. Memory usage is similarly small: because no gradients or optimizer states are stored, GeoDetect only keeps a minibatch of embeddings and an $n×n$ distance matrix, which is negligible compared to model weights.
>
> Also, GeoDetect processes embeddings one minibatch at a time, so runtime grows linearly with the dataset size, avoiding the quadratic $N^2$ cost of full-dataset $k$-NN. For a minibatch of size $n$ and embedding dimension $d$, the computational cost is $O(n^2.d)$ for k-NN/KDE/LID and $O(n.d^2)$ for Mahalanobis, which are inexpensive in practice.
>
> By contrast, training-based robustness methods are significantly more costly. A linear probe [1] requires $O(N⋅C⋅d)$ operations for forward/backward passes of a $C×d$ classifier over $N$ samples, while adapter [2] and prompt-tuning [3] methods incur an additional $O(T⋅d^2)$ cost from backpropagating through $d×d$ projection or prompt-update layers over $T$ optimization steps. GeoDetect avoids all such backward-pass costs by operating entirely in frozen-encoder inference mode.
>
> [1] Radford, Alec, et al. "Learning transferable visual models from natural language supervision." International Conference on Machine Learning (ICML), 2021.
>
> [2] Zhou, Kaiyang, et al. "Learning to prompt for vision-language models." International Journal of Computer Vision 130.9 (2022): 2337-2348.
>
> [3] Gao, Peng, et al. "Clip-adapter: Better vision-language models with feature adapters." International Journal of Computer Vision 132.2 (2024): 581-59
>
> > **W2.** Multimodal attacks evaluation
>
> We clarify that GeoDetect has already been evaluated under joint image-text adversarial attacks. In particular, we include cross-modal attacks such as Co-Attack (Section 4) and SGA (Appendix D.2), which simultaneously perturb both the image and text inputs. The details of these attacks are provided in Appendix A.1. These attacks directly target the multimodal encoder or the image–text matching objective and represent complex cross-modal adversarial scenarios. As reported in our experiments for both zero-shot classification and image–text retrieval, GeoDetect remains robust even when the text modality is adversarially perturbed together with the image.
>
> > **W3.** Expected findings
>
> To the best of our knowledge, this is the first work to analyze the detectability of adversarial examples in VLPs both theoretically and empirically, across multiple architectures and across both zero-shot classification and retrieval. This addresses a clear gap in prior literature [1-3], which focuses on unimodal classifiers, assumes access to logits, or lacks multimodal generality.
>
> Also, part of our contribution lies in formally connecting CLIP/VLP geometry to distance-based adversarial detection. In Section 3.2.1, we analyze the anisotropy of VLP embeddings and show that multiple VLP families exhibit strong covariance misalignment. Building on this, Section 3.2.2 (Lemmas 3.3, 3.4, and Theorem 3.5) establishes that adversarial perturbations increase their expected distance to clean samples under anisotropic covariance, providing a principled explanation for why geometric signals should separate adversarial from clean embeddings. Finally, Appendix B.5 derives how this off-manifold deviation induces systematic and measurable changes across all four geometric metrics.
>
> We believe that demonstrating effective detectability using theoretically grounded geometric signals in the VLP setting, together with empirical validation, offers valuable insight for the community and meaningfully extends prior distance-based detection methods to a new and practically important domain.

---

> ### Comment · Reviewer_zP31 · 2025-11-26
> **response to the authors**
>
> Thanks for the rebuttal. However, I still think the conclusion that adversarial examples tend to move off the data manifold is not surprising.

---

### Official Review · Reviewer_mPpr · 2025-10-24

**Soundness:** 3
**Presentation:** 3
**Contribution:** 2
**Rating:** 6
**Confidence:** 4

**Summary:**

The authors propose a theoretical and practical framework for detecting adversarial examples in VLPs. The authors first show that VLP embedding spaces are anisotropic (i.e., that data representations cluster unevenly along certain dimensions). They prove that this anisotropy causes adversarial perturbations to push samples off the data manifold, resulting in increased geometric distances from clean examples. Building on this insight, the paper introduces GeoDetect, a lightweight, model-agnostic detection method that computes geometric scores (Local Intrinsic Dimensionality, k-NN distance, Mahalanobis distance, and Kernel Density Estimation) on image or multimodal embeddings to distinguish clean and adversarial inputs.

**Strengths:**

- The authors provide a theoretical foundation for adversarial detection in VLPs by linking embedding-space anisotropy to off-manifold deviations under adversarial perturbations. This insight unifies and generalizes previous intuition from unimodal settings.
- Based on this foundation, they introduced GeoDetect, a simple yet powerful and model-agnostic detection framework that uses classical geometric metrics to achieve near-perfect detection performance across diverse architectures and tasks without retraining or fine-tuning.
- They demonstrate strong empirical generalization and robustness, showing consistent detection accuracy across VLP families and downstream tasks.
- They also demonstrate their method's robustness under adaptive attack scenarios.

**Weaknesses:**

- Their fundamental claim that adversarial examples lie off the manifold in latent space is actually not new, having been demonstrated in prior unimodal contexts. The authors have actually done well to discuss this, but as this is foundational to their defense design the novelty is limited (originality claim remains strong however).

- The anisotropy analysis, though central to the theoretical argument, remains largely descriptive. While measures such as $(I_1, I_2)$, and effective rank support anisotropy qualitatively, the paper never quantifies how anisotropy correlates with detection performance or adversarial vulnerability across architectures. This weakens the causal link between the theoretical foundation and the empirical effectiveness of GeoDetect.

- The method’s scalability and computational efficiency are underexplored. Geometric metrics such as k-NN, Mahalanobis, and KDE in practice scale poorly with embedding dimensionality and dataset size. The paper claims GeoDetect is “lightweight,” but offers no analysis of runtime complexity, memory footprint, or potential bottlenecks for real-time applications.

- The paper focuses primarily on standard, gradient-based PGD-style attacks. It lacks experiments on modern, semantically aligned or diffusion-based attacks that perturb both modalities coherently. Without this, it’s unclear if the geometric signal exploited by GeoDetect generalizes to more adaptive or distribution-preserving adversaries.

- While the authors have done most things right in the paper, they omitted analysis on failure cases and sensitivity to model variance. While results show near-perfect AUC scores, no discussion is offered on outliers, false positives, or potential degradation in low-sample or highly anisotropic conditions (e.g., small-scale VLPs or different pre-training datasets). This absence makes the method’s reliability across unseen regimes uncertain, especially given that embedding geometry can vary substantially between pre-training objectives.

**Questions:**

Check weaknesses above

---

> ### Author Response · Authors · 2025-11-21
> **Response to Reviewer mPpr (1/2)**
>
> Thank you for taking the time to review our paper and providing valuable feedback. Below, you will find our responses to your questions:
>
> > **W1.** Fundamental claim
>
> To the best of our knowledge, this is the first study to evaluate the detectability of adversarial images in VLPs both theoretically and empirically across multiple tasks, and is generalizable to different downstream tasks, addressing a notable gap in the existing literature.
>
> While the high-level intuition has appeared in unimodal contexts, our contribution is to formally connect VLP embedding structure to off-manifold adversarial behavior and to demonstrate that this relationship holds in multimodal settings where the geometry differs from unimodal models. In Section 3.2.2, we establish that under the anisotropic covariance structure measured in Section 3.2.1, adversarial perturbations provably increase their expected distance from clean samples across geometric metrics. This theoretical link, specific to multimodal embeddings, has not been shown in prior work.
>
> We believe that demonstrating effective detectability using existing methods in the context of a new problem provides valuable insights to the community. Our rigorous experiments across both zero-shot classification (Tables 1, 15) and image-text retrieval (Table 2), together with evaluations spanning simple image attacks, multimodal attacks such as Co-Attack and SGA (Table 11), and strong adaptive attacks (Tables 13, 14), consistently validate this phenomenon. We view this empirical evidence, combined with our theoretical analysis, as a contribution that is as meaningful as introducing a new detection method.
>
> > **W2.** Anisotropy analysis connection
>
> We would like to clarify a possible misunderstanding that we do not claim that adversarial embeddings are anisotropic. Motivated by prior work on the anisotropic nature of CLIP’s embedding space [3, 4], our analysis begins by confirming that this property is not unique to CLIP. Specifically, we show in Section 3.2.1 and Appendix B.1 that the embedding spaces of other VLPs are also inherently anisotropic. In Assumption 3.1, we state that the anisotropic property applies only to the covariance of clean embeddings. This assumption is then used in Lemma 3.4, which shows that adversarial perturbations, which tend to suppress the tangent components and amplify the normal ones, push the embeddings off the manifold. Therefore, our use of anisotropy refers only to the geometry of clean embeddings and is independent of any specific attack, making the central claim attack-agnostic.
>
> >**W3.** Scalability and computational efficiency
>
> GeoDetect processes embeddings one minibatch at a time, so runtime grows linearly with the dataset size, avoiding the quadratic $N^2$ cost of full-dataset $k$-NN. For a minibatch of size $n$ and embedding dimension $d$, the computational cost is $O(n^2.d)$ for k-NN/KDE/LID and $O(n.d^2)$ for Mahalanobis, which are inexpensive in practice.
>
> By contrast, training-based robustness methods are significantly more costly. A linear probe [1] requires $O(N⋅C⋅d)$ operations for forward/backward passes of a $C×d$ classifier over $N$ samples, while adapter [2] and prompt-tuning [3] methods incur an additional $O(T⋅d^2)$ cost from backpropagating through $d×d$ projection or prompt-update layers over $T$ optimization steps. GeoDetect avoids all such backward-pass costs by operating entirely in frozen-encoder inference mode.
>
> Empirically, Appendix A.4 and Table 3 show that all metrics except LID require only 30–100 seconds, and LID takes ~9 minutes for a full CIFAR-10 run on a single H100 GPU. Memory usage is similarly small: because no gradients or optimizer states are stored, GeoDetect only keeps a minibatch of embeddings and an $n×n$ distance matrix, which is negligible compared to model weights.
>
> [1] Radford, Alec, et al. "Learning transferable visual models from natural language supervision." International Conference on Machine Learning (ICML), 2021.
>
> [2] Zhou, Kaiyang, et al. "Learning to prompt for vision-language models." International Journal of Computer Vision 130.9 (2022): 2337-2348.
>
> [3] Gao, Peng, et al. "Clip-adapter: Better vision-language models with feature adapters." International Journal of Computer Vision 132.2 (2024): 581-595.
>
> > **W4.** Multimodal attacks
>
> We clarify that GeoDetect has already been evaluated under joint image-text adversarial attacks. In particular, we have included cross-modal attacks such as Co-Attack (Section 4) and SGA (Appendix D.2), which simultaneously perturb both the image and text inputs. The details of these attacks are provided in Appendix A.1. These attacks directly target the multimodal encoder or the image–text matching objective and represent complex cross-modal adversarial scenarios. As reported in our experiments for both zero-shot classification and image–text retrieval, GeoDetect remains robust even when the text modality is adversarially perturbed together with the image.

---

> > ### Author Response · Authors · 2025-11-21
> > **Response to Reviewer mPpr (2/2)**
> >
> > >**W5.** Sensitivity analysis and failure cases
> >
> > We already mentioned most of the concerns. First, we report FPR95 together with AUC across models, datasets, and attacks, showing that near-perfect AUC does not come from trivial thresholds and that false positives remain low even under strong attacks.
> >
> > Second, Appendix C presents a sensitivity analysis over locality k, batch size, layer choice, and the number of available samples, where performance remains stable even when NNN is significantly reduced, indicating robustness in low-sample regimes.
> >
> > Third, Section 3.2.1 and Appendix B.7 provide anisotropy and off-manifold diagnostics (PCA/SVD tail energy, KL along principal directions) that explicitly characterize how adversarial embeddings move away from the clean manifold under highly anisotropic conditions across four VLP families.
> >
> > Taken together, these results show that GeoDetect is stable across model variance, anisotropy levels, and sample sizes.

---

> > > ### Comment · Reviewer_mPpr · 2025-11-27
> > >
> > > I thank the authors for their thoughtful and detailed response. They have clearly engaged with the concerns raised and have clarified several points that were previously unclear. However, while the theoretical analysis is well-presented, I still do not find novelty in the main approach, as I stated in my original assessment. The authors might be overstating the theoretical originality, so I would advise them to avoid this. In addition, although the computational analysis is now clearer, the runtime and memory costs remain comparable to or in some cases higher than existing inference-time VLM detection methods, which somewhat weakens the “lightweight” claim. While I remain confident in the technical soundness and practical relevance of the method, the lack of conceptual novelty would be my main reason to maintain my current ratings. Good job so far, and good luck!

---

### Official Review · Reviewer_XXVv · 2025-10-27

**Soundness:** 2
**Presentation:** 3
**Contribution:** 2
**Rating:** 4
**Confidence:** 4

**Summary:**

GeoDetect is a geometry-based adversarial detection framework for vision–language models (VLPs).
It leverages the anisotropy of multimodal embeddings and formulates a measurable *expected distance gap* to separate clean and adversarial samples through geometric scores (LID, k-NN, Mahalanobis, KDE).
The approach is model- and task-agnostic, showing strong results on zero-shot classification and image-text retrieval, though it still relies on key assumptions and limited attack coverage.

**Strengths:**

* **Clear theoretical motivation.** The paper clearly connects VLP anisotropy and “off-manifold” adversarial behavior, and mathematically supports using geometric scores (LID, k-NN, Mahalanobis, KDE) as detection criteria.
* **Model- and task-independent.**  The framework works without access to task heads or logits, transferring naturally between zero-shot classification and cross-modal retrieval.
* **Broad experimental setup.** The authors evaluate single-modal, multimodal, and text-perturbation scenarios, and report robustness to several adaptive attacks and different attack backbones.

**Weaknesses:**

* **Dependence on anisotropy assumptions.** The method’s validity heavily relies on the anisotropy hypothesis. It remains uncertain whether this still holds under stronger attacks such as **M-Attack** or **SA-AET (Semantic-Aligned Adversarial Evolution Triangle)**
, which might deliberately align adversarial embeddings with the original manifold.
* **Limited baselines.** Only one prior baseline from 2022 is compared, which is insufficient to demonstrate progress against more recent adversarial detection methods for VLPs.
* **Reference set dependence.** The method requires a clean sample pool for computing geometric metrics, but the paper does not explain how to construct, maintain, or recalibrate this reference set under distribution shift or contamination.
* **Layer-specific sensitivity.** Although the paper notes that LID is most effective at multimodal layers, it also shows that in certain ALBEF multimodal attacks, k-NN/KDE degrade substantially. A systematic analysis or automatic layer-selection strategy is missing.

**Questions:**

The chosen attacks and baselines are overly simplistic; the current experimental setup is insufficient to demonstrate the method's effectiveness under stronger or more realistic threats.

---

> ### Author Response · Authors · 2025-11-21
> **Response to Reviewer XXVv (1/2)**
>
> We sincerely appreciate the reviewer’s time and constructive feedback. We would like to clarify the following points that may have been misinterpreted.
>
> > **W1.** Dependence on anisotropy assumptions and evaluation of stronger attacks
>
> We would like to clarify a possible misunderstanding that we do not claim that adversarial embeddings are anisotropic. Motivated by prior work on the anisotropic nature of CLIP’s embedding space [1, 2], our analysis begins by confirming that this property is not unique to CLIP. Specifically, we show in Section 3.2.1 and Appendix B.1 that the embedding spaces of other VLPs are also inherently anisotropic. In Assumption 3.1, we state that the anisotropic property applies only to the covariance of clean embeddings. This assumption is then used in Lemma 3.4, which shows that adversarial perturbations, which tend to suppress the tangent components and amplify the normal ones, push the embeddings off the manifold. Therefore, our use of anisotropy refers only to the geometry of clean embeddings and is independent of any specific attack, making the central claim attack-agnostic.
>
> We also evaluated GeoDetect_LID under the newest attack, SA-AET [3], using the Flickr30k dataset, and the results in the table below show that its performance remains stable.
>
> | Model       | Co-attack | SGA    | SA-AET |
> |-------------|-----------|--------|--------|
> | CLIP_RN50   | 98.90     | 88.18  | 78.73  |
> | CLIP_ViT    | 96.55     | 85.47  | 72.75  |
> | ALBEF       | 93.80     | 73.75  | 69.38  |
> | TCL         | 90.76     | 70.12  | 64.11  |
>
> [1] Liang, Victor Weixin, et al. "Mind the gap: Understanding the modality gap in multi-modal contrastive representation learning." Advances in Neural Information Processing Systems (NeurIPS), 2022.
>
> [2] Levi, Meir Yossef, and Guy Gilboa. "The double-ellipsoid geometry of clip." arXiv preprint arXiv:2411.14517, 2024.
>
> [3] Jia, Xiaojun, et al. "Semantic-aligned adversarial evolution triangle for high-transferability vision-language attack." IEEE Transactions on Pattern Analysis and Machine Intelligence, 2025.
>
>
> >**W2.** Limited Baselines
>
> We already compared our method with the newest baseline PIP [1]  in Appendix E.2, and our results show that GeoDetect consistently surpasses PIP. However, PIP is fundamentally task-specific to VQA and relies on question-conditioned cross-attention maps, which makes it incompatible with dual-encoder architectures and unusable for general zero-shot classification or image–text retrieval. To the best of our knowledge, MCM remains the only prior method that is applicable across multiple tasks in the zero-shot, frozen-model setting without any tuning, and therefore serves as the most appropriate general baseline.  If the reviewer has additional suggestions for alternative baselines, we would be happy to include them in our comparison.
>
> [1] Zhang, Yudong, et al. "Pip: Detecting adversarial examples in large vision-language models via attention patterns of irrelevant probe questions." ACM International Conference on Multimedia, 2024.
>
> > **W3.** Reference set dependence
>
> We would like to clarify that GeoDetect does not rely on a fixed global clean reference set. As mentioned in the paper (lines 192-194), all geometric metrics are computed on minibatch-local neighborhoods, so the detector only requires embeddings from the current batch rather than a clean pool. This avoids the need to construct or maintain a large reference set and naturally adapts to distribution changes, since neighborhoods are recomputed during inference.
>
> In addition, our sensitivity analysis in Appendix C evaluates robustness with respect to locality, batch size, and the number of available clean samples. Across all these settings, GeoDetect remains stable, further demonstrating that the method does not depend on a carefully maintained reference set.
>
> >**W4.** Layer-specific sensitivity
>
> We describe and justify our layer-selection choices in Appendix A.3, where we follow standard practice in prior work [4] and use the final embedding layer for all geometric metrics except LID. For LID, we explicitly aggregate information from multiple layers, including the multimodal fusion layer, which we show empirically to be the most informative. In addition, Appendix C.3 provides a layer-wise analysis detailing how we selected the layers within the model.
>
> The degradation observed under multimodal attacks on ALBEF is expected and explained in lines 450–45: k-NN/KDE rely on a single multimodal representation, whereas LID incorporates both unimodal and multimodal features, making it more robust to such attacks.
>
> [4] Ma, Xingjun, et al. "Characterizing Adversarial Subspaces Using Local Intrinsic Dimensionality." International Conference on Learning Representations (ICLR), 2018.

---

> ### Author Response · Authors · 2025-11-21
> **Response to Reviewer XXVv (2/2)**
>
> >**Q1.** Simple attacks and baselines
>
> We respectfully disagree that the attack setting is overly simplistic. The related papers in the literature [5-8] use the same attacks as ours, or even simpler ones, for example, PGD-based image-only attacks are just evaluated in [2]. Beyond standard multi-step PGD, we evaluate GeoDetect under
>
> (i) strong joint image–text attacks such as Co-Attack (Main text Section 4.1) and SGA (Appendix D.2) for retrieval (including black-box transfer: Appendix D.2.1), and
>
> (ii) adaptive attacks (Appendix D.3) that explicitly incorporate our geometric scores into the optimization objective, including different-distribution adaptive attacks and the selective-gradient strategy.
>
> Across these stronger and more realistic threat models, GeoDetect remains robust even when the attacker directly targets the detector.
>
> [5] Li, Qin, et al. "Joint Adversarial Purification: Mitigating the Threat of Multimodal Adversarial Examples." Proceedings of the International Conference on Multimedia Retrieval, 2025.
>
> [6] Zhang, Yudong, et al. "Pip: Detecting adversarial examples in large vision-language models via attention patterns of irrelevant probe questions." Proceedings of the 32nd ACM International Conference on Multimedia, 2024.
>
> [7] Wang, Youze, et al. "Boosting Adversarial Robustness of Vision-Language Pre-training Models against Multimodal Adversarial attacks." ICLR 2025 Workshop on Building Trust in Language Models and Applications, 2025.
>
> [8] Zhao, Yunqing, et al. "On evaluating adversarial robustness of large vision-language models." Advances in Neural Information Processing Systems (NeurIPS), 2023.

---

### Official Review · Reviewer_C6a5 · 2025-11-03

**Soundness:** 2
**Presentation:** 3
**Contribution:** 2
**Rating:** 2
**Confidence:** 3

**Summary:**

In this paper, the authors introduce GeoDetect, a lightweight and model-agnostic approach for detecting adversarial examples on vision-language pre-trained models (VLPs), such as CLIP. The authors observe that VLP embeddings are anisotropic, meaning that clean data embeddings lie on a data manifold, and as a consequence adversarial examples move off that manifold. GeoDetect uses simple geometric metrics—like k-NN, Mahalanobis distance, LID, and KDE—to quantify these off-manifold shifts and identify adversarially attacked examples. Experiments on several datasets and attacks show that GeoDetect achieves good adversarial detection performance across architectures and tasks, offering an efficient defense for multimodal models.

**Strengths:**

* The paper addresses a timely and important challenge, detecting adversarial attacks in vision-language pre-trained models, offering a practical contribution to improving the safety of multimodal AI models.

* The approach is simple, efficient, and broadly applicable, requiring no model fine-tuning or task-specific modifications.

* It is clearly written and easy to follow; the main idea of the proposed method is intuitive.

**Weaknesses:**

* one of the main weaknesses of the paper is the limited novelty of the proposed approach. Using embedding-space distances or density-based measures for adversarial detection has been explored in prior works (e.g., [0, 1, 2]), which makes the contribution primarily an adaptation of existing ideas to multimodal settings.

* the method does not account for adaptive attacks specifically designed to keep adversarial examples on the clean data manifold. Such attacks could significantly degrade detection performance, yet this scenario is not evaluated in the experiments.

* moreover, prior studies have already demonstrated that adversarial attacks incorporating distance or manifold regularization terms can successfully bypass similar geometric defenses (e.g., [4, 5]), highlighting the need for a more thorough robustness evaluation.

[1] Ma, Xingjun, et al. "Characterizing adversarial subspaces using local intrinsic dimensionality." arXiv preprint arXiv:1801.02613 (2018).

[2] Lee, Kimin, et al. "A simple unified framework for detecting out-of-distribution samples and adversarial attacks." Advances in neural information processing systems 31 (2018).

[3] Cohen, Gilad, Guillermo Sapiro, and Raja Giryes. "Detecting adversarial samples using influence functions and nearest neighbors." Proceedings of the IEEE/CVF conference on computer vision and pattern recognition. 2020.

[4] Athalye, Anish, Nicholas Carlini, and David Wagner. "Obfuscated gradients give a false sense of security: Circumventing defenses to adversarial examples." International conference on machine learning. PMLR, 2018.

[5] Bryniarski, Oliver, et al. "Evading adversarial example detection defenses with orthogonal projected gradient descent." arXiv preprint arXiv:2106.15023 (2021).

**Questions:**

* how does GeoDetect differ from prior geometric or distance-based adversarial detection methods ([1–3]) beyond applying them to multimodal settings?

* how would the method perform against adaptive attacks designed to keep adversarial examples on the clean data manifold?

* given that prior works ([4, 5]) show such attacks can bypass geometric defenses, how robust is GeoDetect under similar adaptive scenarios?

---

> ### Author Response · Authors · 2025-11-21
> **Response to Reviewer C6a5**
>
> We sincerely appreciate the reviewer’s time and constructive feedback. We would like to clarify the following points that may have been misinterpreted.
>
> > **Q1.** Difference between GeoDetect and prior methods
>
> To the best of our knowledge, this is the first work to analyze the detectability of adversarial examples in VLPs both theoretically and empirically, across multiple architectures and across both zero-shot classification and retrieval tasks. This addresses a clear gap in prior literature [1-3], which focuses on unimodal classifiers, assumes access to logits, or lacks multimodal generality.
>
> Also, part of our contribution lies in formally connecting VLPs' geometry to distance-based adversarial detection. In Section 3.2.1, we analyze the anisotropy of VLP embeddings and show that multiple VLPs exhibit covariance misalignment. Building on this, Section 3.2.2 (Lemmas 3.3, 3.4, and Theorem 3.5) establishes that adversarial perturbations increase their expected distance to clean samples under anisotropic covariance, providing a principled explanation for why geometric signals should separate adversarial from clean embeddings. Finally, Appendix B.5 derives how this off-manifold deviation causes measurable changes across all geometric metrics.
>
> Finally, we believe that demonstrating effective detectability using theoretically grounded geometric signals in the VLP settings, together with empirical validation, offers valuable insight for the community and meaningfully extends prior distance-based detection methods to a new and practically important domain.
>
> [1] Ma, Xingjun, et al. "Characterizing Adversarial Subspaces Using Local Intrinsic Dimensionality." International Conference on Learning Representations (ICLR), 2018.
>
> [2] Lee, Kimin, et al. "A simple unified framework for detecting out-of-distribution samples and adversarial attacks." Advances in neural information processing systems (NeurIPS), 2018.
>
> [3] Cohen, Gilad, et al. "Detecting adversarial samples using influence functions and nearest neighbors." Proceedings of the IEEE/CVF conference on computer vision and pattern recognition (CVPR), 2020.
>
> > **Q2 & Q3.** Adaptive attacks [4,5] evaluation
>
> We already evaluated both strong adaptive attacks proposed in [4,5] in detail in Appendix D.3, and our results show that GeoDetect remains robust even when the attacker directly optimizes against the geometric metrics.
>
> [4] Athalye, Anish, Nicholas Carlini, and David Wagner. "Obfuscated gradients give a false sense of security: Circumventing defenses to adversarial examples." International conference on machine learning (ICML), 2018.
>
> [5] Bryniarski, Oliver, et al. "Evading adversarial example detection defenses with orthogonal projected gradient descent." arXiv preprint arXiv:2106.15023 (2021).

---

### Meta-Review · Area_Chair_3YiP · 2026-01-07

**Summary:**

The reviewers acknowledged the paper's contribution to the relatively unexplored area of adversarial detection in Vision-Language Pre-trained models (VLPs). The core strength of the work lies in its geometric approach, which utilizes the anisotropic structure of VLP embedding spaces to identify adversarial examples. Reviewers generally found the theoretical analysis—linking off-manifold deviations to systematic changes in geometric metrics—to be a principled and interesting way to differentiate clean and adversarial inputs. The empirical performance across multiple VLP architectures also served as a strong point of validation.

However, the primary concern raised during the review process was the degree of conceptual novelty. Reviewer zP31, in particular, argued that the finding that adversarial examples move off the data manifold is a well-established phenomenon in single-modality settings, suggesting that the extension to VLPs might be incremental. There were also initial requests for more rigorous comparisons against state-of-the-art distance-based detectors and a clearer explanation of how the image and text modalities interact geometrically. While the authors' rebuttal clarified the systematic nature of these changes across four distinct geometric metrics, the debate over the "obviousness" of the core manifold-deviation premise remained the central point of contention in the final decision.

The AC also checked the paper and went through the review comments.

**Reviewer Concerns:**

Reviewers praised the exploration of adversarial detection in VLPs, specifically the use of the anisotropic embedding space to identify off-manifold adversarial examples. The theoretical analysis linking manifold deviation to geometric metrics was considered principled, and empirical results across different VLP architectures were strong.

The primary debate centered on conceptual novelty. Reviewer zP31 argued that the "off-manifold" nature of adversarial examples is a well-known phenomenon in single-modality settings, viewing the extension to VLPs as incremental. While the authors' rebuttal clarified the systematic nature of these changes and provided more rigorous comparisons, the critical concerns were still not addressed.

**Reviewer Scores:**

C6a5, 2 (Reject) ,  Remains 2

XXVv, 4 (Marginally Below) ,Remains 4

mPpr, 6 (Marginally Above) , ,Remains 6

zP31, 4 (Marginally Below) ,Remains 4

---

### Decision · Program_Chairs · 2026-01-26

Reject